# Robust Deep Reinforcement Learning against Adversarial Behavior Manipulation

## Abstract

This study investigates the robustness of deep reinforcement learning agents against targeted attacks that aim to manipulate the victim's behavior through adversarial interventions in state observations. While several methods for such targeted manipulation attacks have been proposed, they all require white-box access to the victim's policy, and some rely on environment-specific heuristics. Furthermore, no defense method has been proposed to counter these attacks. To address this, we propose a novel targeted attack method for manipulating the victim, which does not depend on environmental heuristics and applies in black-box and no-box settings. Additionally, we introduce a defense strategy against these attacks. Our theoretical analysis proves that the sensitivity of a policy's action output to state changes affects the defense performance and that the earlier in the trajectory, the greater the effect. Based on this insight, we introduce a time-discounted regularization as a countermeasure for such behavior targeted attacks, which helps to improve robustness against attacks while maintaining task performance in the absence of attacks. Empirical evaluations demonstrate that our proposed attack method outperforms baseline attack methods. Furthermore, our defense strategy shows superior robustness against existing defense methods designed for untargeted attacks.

## 1 Introduction

Applications of Deep Reinforcement Learning (DRL) have grown significantly in recent years (Berner et al., 2019; Kiran et al., 2021; Ouyang et al., 2022). However, trained DRL agents remain vulnerable to adversarial attacks (Huang et al., 2017). While numerous studies have explored adversarial attacks and defenses in DRL agents (Pattanaik et al., 2018; Zhang et al., 2020b; Wu et al., 2022), much of the attention has been pained to untargeted attacks, where the adversary aims to minimize the cumulative reward the victim earns from the environment. In real-world scenarios, the adversary's goal can be more specific—manipulating the victim's behavior to achieve malicious outcomes. For example, in autonomous driving, an adversary could control the route or speed to cause a collision or traffic jam under certain conditions. These behavior-manipulating attacks are more precise and sophisticated than untargeted ones, presenting a stealthier and more severe threat. In this work, we define such attacks as behavior-targeted attacks (Figure 1). One of the key objectives of this study is to develop a methodology to attain behavior-targeted attacks, assuming that the adversary can intervene in the victim's observations and induce false observations of the environment. It is important to note that our focus is on attacking trained victim agents, which differs from poisoning attacks that occur during the victim's training process.

Prior works proposing behavior-targeted attacks face several limitations. Hussenot et al. (2020) introduced an attack method that precomputes universal adversarial perturbations for each action in advance, forcing the victim to choose actions consistent with the adversary's specified policy at each step. However, this approach becomes impractical when the action space is large or continuous, limiting its applicability in more complex tasks. Boloor et al. (2020) developed an attack method specifically for autonomous driving agents. This method, however, relies on a heuristic objective function that requires detailed knowledge of the autonomous driving task, resulting in a lack of generalizability across other tasks. A further limitation common to both methods is their dependence on white-box settings that assume full access to the victim's policy. In many real-world scenarios,

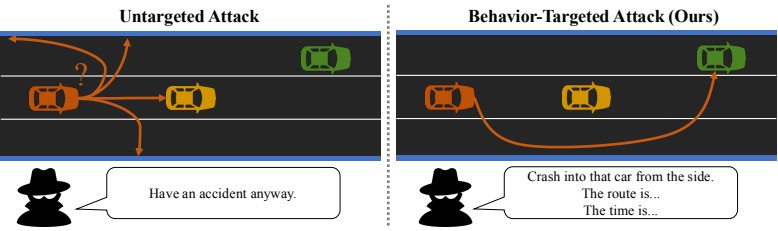

Figure 1: Overview of behavior-targeted attack.

an adversary is unlikely to have full knowledge of the victim's policies, making this assumption severely limits its practical application.

In this work, we propose a novel behavior-targeted attack, Behavior Imitation Attack (BIA), which is applicable even under limited access to the victim's policy (i.e., black-box and no-box settings) and does not require environment-specific heuristics. Specifically, we define the adversary's objective as forcing the victim to imitate a policy specified by the adversary, referred to as the target policy. As demonstrated in Section 7, a naive approach that attempts to match the victim's action with the target policy's action at each step is ineffective. This result is because when the adversary's intervention capability is limited, there are often no false states in which the victim's behavior fully aligns with the target policy. Instead, our approach aims to guide the victim to imitate the overall behavior rather than matching individual actions at each step. Consequently, our attack can achieve overall behavioral alignment even in situations where perfect action matching is not possible at each step.

Another goal of this study is to develop a countermeasure for the behavior-targeted attack. We present the first robust training framework, Time-Discounted Reglarization Training (TDRT), for such attacks as a defense method. Our theoretical analysis reveals two important insights: (1) suppressing the sensitivity of the policy's action outputs to state changes enhances robustness against attacks, and (2) achieving less sensitivity during the early stages of a trajectory significantly improves overall robustness. Through this analysis, we introduce a time-discounted regularization approach, where stronger regularization is applied during the critical early stages of the trajectory and gradually weakened in the later, less critical stages.

This method is particularly effective against attacks that do not reduce rewards much but alter the resulting behavior significantly. This is especially important for situations where a change in behavior does not correlate to the reward.Since defense methods designed for untargeted attacks cannot deal with this type of attack, this regularization strategy is effective for behavior-targeted attacks. Furthermore, time-discounted regularization mitigates performance degradation without compromising robustness compared to regularization without time discounting. Consequently, time discounting improves the trade-off between robustness against attacks and performance under non-attack conditions. To the best of our knowledge, no defense mechanism has been considered for the behavior-targeted attack.

Our primary contributions can be summarized as follows:

- We formulate the concept of behavior-targeted attack in reinforcement learning and propose the Behavior Imitation Attack (BIA), an attack algorithm in this setting. BIA does not require environment-specific heuristics and is applicable in both black-box/no-box settings.

- We propose a robust training framework, Time-Discounted Regularization Training (TDRT). TDRT is robust against attacks that do not significantly reduce rewards but greatly alter behavior. Existing defenses designed for untargeted attacks are ineffective against such threats. Furthermore, time-discounted regularization mitigates performance degradation without compromising robustness, outperforming regularization methods without time discounting. To the best of our knowledge, TDRT is the first defense method specifically designed to counter behavior-targeted attacks.

- We demonstrate the effectiveness of our proposed attack and defense methods using the reinforcement learning benchmark Meta-World (Yu et al., 2020) and MuJoCo environment. Our attack method is more effective than baselines.

## 2 RELATED WORKS

**Untargeted attack and its defense**  Untargeted attacks aim to minimize the victim's cumulative reward. Gradient-based methods, which require white-box access to the victim's policy, have been proposed Huang et al. (2017); Pattanaik et al. (2018); Gleave et al. (2020). One common defense method against untargeted attacks is policy smoothing. Shen et al. (2020) and Zhang et al. (2020b) introduced regularization terms to smooth the policy, demonstrating increased robustness. However, unlike our approach, these methods do not incorporate time discounting. Additionally, adversarial training has been proposed Zhang et al. (2021); Oikarinen et al. (2021); Sun et al. (2022); Liang et al. (2022). While these methods demonstrated high levels of robustness, Korkmaz (2021; 2023) pointed out that agents trained to be adversarial robust are still vulnerable to attacks that were not anticipated during training. Liang et al. (2024) introduced a new concept called temporally-coupled perturbations, where consecutive perturbations are constrained. Belaire et al. (2024) proposed a novel defense method based on regret minimization. Liu et al. (2024) have proposed PROTECTED, which is effective not only against worst-case attacks but also against more general forms of attack. Another approach is certified defense (Wu et al., 2022; Kumar et al., 2022; Mu et al., 2024; Sun et al., 2024). These methods guarantee a lower bound on the reward obtained by that victim under adversarial attacks.

**Targeted attack**  Targeted attacks manipulate the victim to follow a specific direction and can be categorized into two main types: enchanting attacks and behavior-targeted attacks. Enchanting attacks aim to lure the victim to a predetermined final state without specifying the entire behavior trajectory. Lin et al. (2017) and Tretschk et al. (2018) proposed methods to guide the victim to a target state through crafted perturbations. More recently, Ying et al. (2023) introduced an enchanting attack using universal adversarial perturbations. Behavior-targeted attacks, on the other hand, aim for more detailed control over the victim's entire behavior. The introductory section has discussed the works of (Hussenot et al., 2020) and (Boloor et al., 2020) in detail.

**Poisoning attack**  Several studies have proposed poisoning attacks against RL agents that intervene during the victim's training phase (Sun et al., 2021; Rakhsha et al., 2020; Rangi et al., 2022; Xu et al., 2023; Xu & Singh, 2023). These works explored various approaches, from theoretical investigations to practical black-box attacks. However, as our research focuses on attacking trained victims rather than interfering during training, our threat model differs fundamentally from these poisoning attack studies. A comprehensive review of related work is provided in Appendix A.

## 3 PRELIMINARIES

Our threat model supposes that the adversary can make the victim observe false states that are different from the true observation of states. Zhang et al. (2020b) formulated this threat model for the first time as the State-Adversarial Markov Decision Process (SA-MDP) and introduced the optimal untargeted attacks. Our work is built on the SA-MDP to formulate the behavior-targeted attack. In addition, we utilize imitation learning for the realization of our attack goal. In this section, we first introduce the SA-MDP and then imitation learning, which are essential components of our study.

**Notaion.**  We denote the Markov decision process (MDP) as $(\mathcal{S}, \mathcal{A}, R, p, \gamma)$, where $\mathcal{S}$ is the state space, $\mathcal{A}$ is the action space, $R : \mathcal{S} \times \mathcal{A} \to \mathbb{R}$ is the reward function, and $\gamma \in (0, 1)$ is the discount factor. We use $\mathcal{P}(\mathcal{X})$ as the set of all possible probability measures on $\mathcal{X}$. We denote $p : \mathcal{S} \times \mathcal{A} \to \mathcal{P}(\mathcal{S})$ as the transition probability, $\pi : \mathcal{S} \to \mathcal{P}(\mathcal{A})$ as a stationary policy, and $p_0$ as an initial state distribution. The objective of reinforcement learning is to train a policy that maximizes the expected sum of discounted rewards from the environment. When the agent follows policy $\pi$ in MDP $M$, the objective function can be defined as follows:

$$\max J_{\text{RL}}(\pi) \triangleq \mathbb{E}_{\pi}^{M} \left[ R(s, a, s') \right] = \mathbb{E}_{\substack{s_0 \sim p_0(\cdot) \\ a_t \sim \pi(\cdot|s_t) \\ s_{t+1} \sim p(\cdot|s_t, a_t)}} \left[ \sum_{t=0}^{\infty} \gamma^t R(s_t, a_t, s_{t+1}) \right]. \quad (1)$$

### 3.1 STATE-ADVERSARIAL MARKOV DECISION PROCESS (SA-MDP)

We suppose the victim follows a fixed policy $\pi$ in an MDP $M$. To model the situation where an adversary intervenes in the victim's state observations, we use SA-MDP (Zhang et al., 2020b). In the SA-MDP, we introduce an adversarial policy $\nu : \mathcal{S} \rightarrow \mathcal{P}(\mathcal{S})$ that interferes with the victim agent's state observations by making the victim observe a false state $\hat{s} \sim \nu(\cdot|s)$ at each time without altering the true state of the environment.

The SA-MDP is defined as $\hat{M} = (\mathcal{S}, \mathcal{A}, R, \mathcal{B}, p, \gamma)$, where $\mathcal{B}$ is a mapping from the true state $s \in \mathcal{S}$ to false state space $\mathcal{B}(s) \subseteq \mathcal{S}$ that the adversarial policy can choose from. The size of $\mathcal{B}(s)$ indicates the adversary's intervention capability and is typically set in the neighborhood of $s$. The smaller the size of $\mathcal{B}(s)$, the more challenging it becomes to achieve the attack objective.

Figure 2 illustrates the transition process of the SA-MDP at each step. The process is composed of the following three steps: (i) The adversarial policy receives a true state $s$ from the environment and shows a false state $\hat{s} \sim \nu(\cdot|s)$ to the victim. (ii) The victim observes the false state $\hat{s}$ and selects an action $a \sim \pi(\cdot|\hat{s})$ based on this observation. (iii) The environment transitions to the next state $s' \sim p(\cdot|s,a)$ based on the true state $s$ and the action $a$.

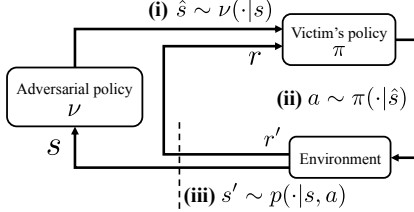

Figure 2: Overview of SA-MDP. The Adversarial policy makes the victim observe the false state from the true state.

### 3.2 IMITATION LEARNING

Imitation learning aims to develop a policy that mimics an expert policy from given demonstrations without requiring rewards from the environment. In our proposed attack method, we leverage imitation learning to train an adversarial policy aimed at manipulating the victim's behavior following the adversary's specification. In this section, we introduce Adversarial Imitation Learning (AIL), a typical imitation learning framework that serves as a key component of our proposal attack method.

**Imitation Learning from Demonstration (ILfD).** Let $\pi_E$ be the expert policy, and $\tau_E$ be the demonstration sampled from $\pi_E$. ILfD is a problem setting where the demonstration $\tau_E$ consists of state-action pairs of length $T$: $\tau_E = \{s_0, a_0, \ldots, s_{T-1}, a_{T-1} \mid a_t \sim \pi_E(\cdot|s_t), s_{t+1} \sim p(\cdot|s_t, a_t)\}$. Without ambiguity, we assume that the expert and learning policies run in the same MDP.

Imitation learning aims to find $\pi$ that minimizes the distance between $\pi$ and $\pi_E$. Ho & Ermon (2016) has shown that this minimization is equivalent to the minimization of the distance between the state-action distributions induced by the policy $\pi$ and the expert policy $\pi_E$ by leveraging the duality of state-action distributions. The state-action distribution $\rho_\pi : \mathcal{S} \times \mathcal{A} \rightarrow [0, 1]$ is defined by the probability of encountering specific state-action pairs when transitioning according to a policy $\pi$:

$$\rho_\pi(s,a) = (1-\gamma)\sum_{t=0}^{\infty}\gamma^t \Pr\left(s_t = s, a_t = a|s_0 \sim p_0(\cdot), a_t \sim \pi(\cdot|s_t), s_{t+1} \sim p(\cdot|s_t, a_t)\right). \quad (2)$$

With this, we formulate the objective of AIL as follows:

$$\min_\pi J_{\text{AIL}}(\pi) \triangleq \mathcal{D}(\rho_\pi, \rho_{\pi_E}), \quad (3)$$

where $\mathcal{D}$ is a distance function that measures the distance between two state-action distributions, for instance, f-divergences such as JS divergence or KL divergence are commonly used for $\mathcal{D}$ (Ho & Ermon, 2016; Kostrikov et al., 2019).

While various algorithms have been proposed to optimize equation 3, GAIL (Ho & Ermon, 2016) introduced the most fundamental optimization algorithm. In GAIL, equation 3 is reformulated as the following GAN-style min-max optimization problem:

$$\min_\pi \max_D J_{\text{GAIL}}(\pi) \triangleq \mathbb{E}_\pi^M[\log(D(s,a))] + \mathbb{E}_{\pi_E}^M[\log(1 - D(s,a))], \quad (4)$$

where $D : \mathcal{S} \times \mathcal{A} \rightarrow [0, 1]$ is a discriminator that tries to distinguish whether a state-action pair is taken from the expert demonstration. The generator is given by $\pi$, which is trained to choose actions such that the discriminator makes incorrect identifications. Training is performed alternately for the generator $\pi$ and the discriminator $D$. In practice, $\mathbb{E}_{\pi_E}^M[\log(1 - D(s,a))]$ is empirically evaluated with demonstration $\tau_E$, instead of evaluation with $\pi_E$.

**Imitation Learning from Observation (ILfO).** ILfO is a problem setting where the demonstration $\tau_E$ consists of only state sequences: $\tau_E = \{s_0, \ldots, s_{T-1} \mid a_t \sim \pi_E(\cdot|s_t), s_{t+1} \sim p(\cdot|s_t, a_t)\}$. Similar to ILfD, the objective of ILfO is the distribution matching, where the distance between state distributions is minimized. Due to the lack of information about the expert's actions, ILfO is known to be empirically more challenging than ILfD.

## 4 THREAT MODEL

In this section, we outline the threat models for the behavior-targeted attack. We assume that the victim follows a fixed policy $\pi$ while it observes false states generated through the adversary's intervention. We define this process as an SA-MDP $M = (\mathcal{S}, \mathcal{A}, R, \mathcal{B}, p, \gamma)$. The adversary's objective is to manipulate the victim into performing a specific behavior desired by the adversary, which is denoted by target policy $\pi_{\text{tgt}}$. To achieve this objective, the adversary trains an adversarial policy $\nu$ that makes the victim observe a false state at each step. In this context, the adversary's influence against the victim is characterized as follows:

**Access to the victim's policy.** We suppose the adversary does not have full access to the victim's policy $\pi$, including the neural network parameters and the training algorithm of the victim's policy. The adversary in the black-box setting can obtain the victim's policy inputs and outputs at each time, specifically the victim's states in the environment and the corresponding actions taken. In contrast, the adversary in the no-box setting can only obtain the victim's state in the environment at each time. The information available to the adversary influences the type of imitation learning the adversary employs for its attack. In the black-box setting, the adversary can use Imitation Learning from Demonstration (ILdD) to train the adversarial policy. In the no-box setting, the adversary needs to use Imitation Learning from Observation (ILfO), which is generally less effective than ILfD in terms of performance.

**Intervention ability on victim's state observations.** We assume that the adversary can alter the victim's state observation from the true state $s$ to the false state $\hat{s} \in \mathcal{B}(s)$. This assumption is common in most adversarial attacks on RL.

## 5 ATTACK METHOD

In this section, we propose a training framework for the adversarial policy, called Behavior Imitation Attack (BIA), as an attack method. In Section 5.1, we define the adversary's objective as a distribution matching approach. However, we cannot directly apply AIL algorithms to learn the adversarial policy due to constraints on the victim's policy. Consequently, in Section 5.2, we provide a theoretical analysis and specific algorithms for learning the adversarial policy.

### 5.1 FORMULATION OF ADVERSARY'S OBJECTIVE

Let $s \in \mathcal{S}$ be the true state. In the SA-MDP framework, the adversarial policy replaces the victim's state observation $s$ with $\hat{s} \sim \nu(\cdot|s)$, and the victim selects the next action by $a \sim \pi(\cdot|\hat{s})$. Consequently, the victim's resulting behavior policy under the adversary's influence is represented by the composite policy $\pi \circ \nu$:

$$\pi \circ \nu(a|s) \triangleq \sum_{\hat{s} \in \mathcal{S}} \nu(\hat{s}|s)\pi(a|\hat{s}). \tag{5}$$

Thus, we define the adversary's objective as a distribution matching problem between $\pi \circ \nu$ and $\pi_{\text{tgt}}$:

$$\min J_{\text{adv}}(\nu) \triangleq \mathcal{D}(\rho_{\pi \circ \nu}, \rho_{\pi_{\text{tgt}}}). \tag{6}$$

To solve this distribution matching problem, we adopt a two-step approach. First, we reformulate this problem of equation 6 into a GAN-style minimax optimization following the GAIL framework in equation 4. Unfortunately, this minimax optimization cannot be solved straightforwardly because it requires optimizing equation 6 w.r.t. the victim's behavior policy $\pi \circ \nu$ as a decision variable. In our attack setting, only the adversarial policy $\nu$ can serve as the decision variable while the victim's policy is fixed. To address this, we introduce a trick that transforms optimization w.r.t. $\pi \circ \nu$ into

---

**Algorithm 1** Behavior Imitation Attack (BIA) in GAIL

---

**Input:** victim's policy $\pi$, initial adversarial policy $\nu_\theta$, initial discriminator $D_\phi$, demonstration $\tau_{\text{tgt}}$, Batch size $B$, learning rate $\alpha$
**Output:** Optimized adversarial policy $\nu_\theta$

1: **for** $n = 0, 1, 2, \ldots$ **do**
2:    $\tau \leftarrow \emptyset$                                           ▷ Initialize empty trajectory
3:    **for** $b = 1$ to $B$ **do**                            ▷ Collect $B$ samples
4:       $\hat{s} \leftarrow \text{Project}(\nu_\theta(s), \mathcal{B}(s))$           ▷ Sample and project state
5:       $a \sim \pi(\cdot|\hat{s})$, $s' \sim p(\cdot|s, a)$       ▷ Sample action and next state
6:       $\tau \leftarrow \tau \cup \{(s, \hat{s}, a, s')\}$, $s \leftarrow s'$
7:    **end for**
8:    $\phi \leftarrow \phi + \alpha\nabla_\phi[\sum_{(s,a)\in\tau} \log D_\phi(s, a) - \sum_{(s,a)\in\tau_{\text{tgt}}} \log(1 - D_\phi(s, a))]$    ▷ Update $D$
9:    **for** $(s, \hat{s}, a, s') \in \tau$ **do**                       ▷ Estimate rewards
10:       $r \leftarrow -\log D_\phi(s, a)$
11:       Update $(s, \hat{s}, a, s')$ to $(s, \hat{s}, a, r, s')$ in $\tau$
12:    **end for**
13:    $\theta \leftarrow \theta + \alpha\nabla_\theta\mathbb{E}_{(s,\hat{s},r)\sim\tau}[r]$              ▷ Update adversarial policy
14: **end for**

---

optimization w.r.t. $\nu$ by Lemma 1 in Section 5.2. Using this, we propose the Behavior Imitation Attack, as described in Algorithm 1, which allows finding adversarial policies for the behavior-targeted attacks.

## 5.2 BEHAVIOR IMITATION ATTACK (BIA)

In this section, we introduce Behavior Imitation Attack (BIA). For simplicity, we consider the attack in the black-box setting by adopting ILfD in the following discussion. However, these arguments are also applicable to the no-box setting with adopting ILfO immediately.

We begin by reformulating equation 6 into the following min-max problem by introducing a discriminator $D$, similarly to the approach used in GAIL, as presented in equation 4:

$$\min_\nu \max_D \mathbb{E}^M_{\pi\circ\nu}[\log(D(s, a))] + \mathbb{E}^M_{\pi_{\text{tgt}}}[\log(1 - D(s, a))]. \tag{7}$$

Focusing on the adversarial policy $\nu$ in equation 7, the adversary solves the following optimization problem:

$$\min_\nu \mathbb{E}^M_{\pi\circ\nu}[\log(D(s, a))] = \max_\nu \mathbb{E}^M_{\pi\circ\nu}[-\log(D(s, a))]. \tag{8}$$

It is important to note that equation 8 is regarded as a reinforcement learning problem aimed at finding a policy $\pi \circ \nu$ that maximizes the cumulative reward, with $-\log(D(s, a))$ serving as a reward function. If the adversary could update the entire $\pi \circ \nu$, equation 8 could be optimized using any RL algorithm. However, since $\pi$ is a fixed policy, directly optimizing equation 8 using standard RL algorithms is infeasible.

Inspired by Lemma 1 from (Zhang et al., 2020b), we present the following lemma to optimize equation 8. This lemma transforms the MDP for equation 8 into a different MDP $\hat{M}$ such that the policy that maximizes equation 8 corresponds to the optimal policy for $\hat{M}$. Since the policy for MDP $\hat{M}$ in equation 11 depends solely on $\nu$ rather than on the composite $\pi \circ \nu$, we can apply any standard RL algorithm to solve this.

**Lemma 1.** *Consider an SA-MDP $M = (\mathcal{S}, \mathcal{A}, R, \mathcal{B}, p, \gamma)$, and let $\pi$ be the victim's policy. Given the reward function $\bar{R}$ specified by the adversary, the reward function $\hat{R}$ and state transition probability $\hat{p}$ are defined as follows:*

$$\hat{R}(s, \hat{s}, s') = \mathbb{E}[\hat{r}|s, \hat{s}, s'] = \begin{cases} \frac{\sum_{a\in\mathcal{A}} \pi(a|\hat{s})p(s'|s,a)\bar{R}(s,a,s')}{\sum_{a\in\mathcal{A}} \pi(a|\hat{s})p(s'|s,a)} & \text{if } \hat{s} \in \mathcal{B}(s) \\ C & \text{otherwise}, \end{cases} \tag{9}$$

$$\hat{p}(s'|s, \hat{s}) = \sum_{a\in\mathcal{A}} \pi(a|\hat{s})p(s'|s, a). \tag{10}$$

*Then, for the MDP $\hat{M} = (\mathcal{S}, \mathcal{S}, \hat{R}, \hat{p}, \gamma)$, the following equality holds:*

$$\nu^* = \arg\max_{\nu} \mathbb{E}_{\nu}^{\hat{M}}[\hat{R}(s, \hat{s}, s')] = \arg\max_{\nu} \mathbb{E}_{\pi \circ \nu}^{M}[\bar{R}(s, a, s')]. \tag{11}$$

The proof is provided in Appendix C.1. With this lemma, we present a practical algorithm for BIA that utilizes GAIL as ILfD to train adversarial policy, as outlined in Algorithm 1. The reward in equation 9 is equivalent to the expected value of $-\log(D(s, a))$ when the adversarial policy chooses false state $\hat{s}$ when the true state is $s$. Thus, in practice, it is sufficient to provide the reward $-\log(D(s, a))$ directly. Also, to stabilize the learning process, we avoid imposing large negative rewards; instead, we limit the range of false states, ensuring that the adversarial policy is enforced to select false states in $\mathcal{B}(s)$ in our experiments.

We emphasize that our behavior-targeted attack method does not necessarily require the adversary to construct the target policy; rather, it functions only with trajectories that are expected to align with the target policy. This characteristic offers a significant advantage in real-world scenarios. Developing the target policy typically involves intricate reward design and policy optimization by reinforcement learning, which requires interaction with the environment and can be extremely costly. In contrast, collecting trajectories can be achieved by reproducing specified behaviors for a few episodes, which is less costly compared to constructing the target policy itself. Section F.2 provides a more detailed analysis of the relationship between trajectories and attack performance.

In addition, our attack method does not depend on environmental heuristics because it is formulated only with the properties of MDPs. Also, since our method does not require direct access to the victim's policy, our method is applicable in the black-box setting when ILfD is adopted and no-box settings when ILfO is adopted for imitation learning.

As a supplement, we prove that the optimization in equation 6 is equivalent to minimizing the distance between policies in Appendix B. This proof ensures that, in our problem setting as well, the distribution matching approach rigorously performs imitation of the target policy.

# 6 DEFENSE METHOD

In this section, we propose a robust training framework against the behavior-targeted attack, Time-Discounted Regularizations Training (TDRT). In Section 6.1, we formulate the defender's objective. In Section 6.2, we provide a theoretical analysis of the objective function and, based on this analysis, introduce a time-discounted regularization term.

## 6.1 FORMULATION OF DEFENDER'S OBJECTIVE

Since the defender has no access to the target policy specified by the adversary, the defender cannot directly maximize the adversary's objective function as given in equation 6 for defense. Instead, we redefine the defender's objective as minimizing the change in the state-action distribution caused by the worst-case adversarial policy $\nu$:

$$\min J_{\text{def}}(\pi) = -J_{\text{RL}}(\pi) + \lambda \max_{\nu} D_{\text{KL}}(\rho_{\pi} || \rho_{\pi \circ \nu}), \tag{12}$$

where $\lambda$ is a hyperparameter that adjusts the trade-off between performance and robustness, and $D_{\text{KL}}$ is the KL divergence. The worst-case adversarial policy $\nu$ is designed to maximize the difference in state-action distributions, serving as a substitute for the inaccessible target policy during the defense training. The distance between the state-action distributions of $\pi$ and $\pi \circ \nu$ indicates the extent to which the victim's behavior has changed under the worst-case attack. The smaller this divergence, the more robust $\pi$ is against behavior-targeted attacks. In other words, equation 12 seeks to develop a robust policy $\pi$ that minimizes changes in the victim's behavior even under the worst-case adversarial policy.

Equation 12 is designed to be robust against behavior-targeted attacks that existing defense methods designed for untargeted attacks cannot mitigate. Existing defenses against untargeted attacks commonly aim to minimize the reduction in cumulative reward caused by attacks. However, policies trained under such objectives may remain vulnerable to attacks that, while not significantly affecting cumulative rewards, change the agent's behavior significantly. In contrast, policies trained with equation 12 can be robust against these behavior-targeted attacks.

## 6.2 TIME-DISCOUNTED REGULARIZATION TRAINING (TDRT)

In this section, we propose a robust reinforcement learning framework, Time-Discounted Regularization Training (TDRT). Directly optimizing equation 12 is challenging as it requires numerous interactions to estimate the state-action distributions. Instead, we prove that equation 12 is bounded by the change in the policy's action distribution in response to state changes:

**Theorem 1.** *Let the discount factor be* $\gamma \in (0,1)$. *For state-action distributions* $\rho_\pi$ *and* $\rho_{\pi \circ \nu}$ *corresponding to policies* $\pi$ *and* $\pi \circ \nu$, *the following inequality holds:*

$$D_{KL}(\rho_\pi || \rho_{\pi \circ \nu}) \le \frac{1}{1-\gamma} \sum_{t=0}^{\infty} \gamma^t \mathbb{E}_{s \sim d_\pi^t} [D_{KL}(\pi(\cdot|s) || \pi \circ \nu(\cdot|s))], \tag{13}$$

*where* $d_\pi^t(s) = \Pr(s_t = s | \pi)$ *represents the state distribution of* $\pi$ *at time* $t$.

The proof is provided in Appendix C.2. In the proof, we bound the state-action distribution using the log-sum inequality. Theorem 1 shows that the smaller the policy's action distribution change in response to state changes, the less the victim's behavior is affected by attacks. It also reveals that the sensitivity of action outputs in the early stages of the trajectory has a greater impact on changes in the victim's behavior. This implies that maintaining low sensitivity of action outputs in the earlier stage of the trajectory can enhance the policy's robustness more against behavior-targeted attacks.

Based on Theorem 1, we introduce a discounted regularization term to more strongly constrain the policy's action distribution at an earlier time. Let $B = \{s_t\}_{t=1}^{N}$ be a mini-batch consisting of states $s_t$ at time $t$. We then redefine the defender's objective by:

$$J_{\text{def}}(\pi) = -J_{\text{RL}}(\pi) + \lambda \max_{\nu} \sum_{s_t \in B} \gamma^t D_{KL}(\pi(\cdot|s_t) || \pi \circ \nu(\cdot|s_t)), \tag{14}$$

We provide the full algorithm for robust training with TDRT when using PPO as the optimizer in Algorithm 2. Note that calculating equation 14 requires the time information of each state. In our implementation, we add time as the last dimension of the state. This time information is used only to calculate the regularization term and is not included in the policy input.

## 7 EXPERIMENTS

In this section, we empirically evaluate the performance of our proposed method using Meta-World (Yu et al., 2020), a benchmark that simulates robotic arm manipulation. We assess the following three aspects: (i) Attack Performance: Can the adversarial policy learned through BIA effectively manipulate the victim's behavior? (ii) Robustness: Are policies learned through TDRT more robust against behavior-targeted attacks than conventional defense methods designed for untargeted attacks? (iii) Trade-off: Can the time-discounted regularization term in TDRT suppress performance degradation while maintaining robustness better than regularization without time discounting? We conducted additional experiments in the MuJoCo environment in Appendix F. Detailed descriptions of experiments are provided in Appendix G.

### 7.1 SET UP

**Adversary's Objective and Setting** In real-world attack scenarios, an adversary would aim to force a victim, trained to act safely to avoid harming people, into performing harmful actions. Therefore, we set the adversary's goal to force the victim to complete tasks that are the exact opposite of what it has been trained to do. To evaluate this, we designed experiments on four tasks featuring opposing goals: {window-close, window-open} and {drawer-close, drawer-open}.

Following the prior works (Zhang et al., 2021; Sun et al., 2022), we constrain the set of adversarial states $\mathcal{B}(s)$ using the $L_\infty$ norm: $\mathcal{B}(s) \triangleq \{\hat{s} \mid \|\hat{s} - s\|_\infty \le \epsilon\}$, where $\epsilon$ represents the attack budget. All experiments are performed with $\epsilon = 0.3$. States are standardized across all tasks, with standardization coefficients calculated during the training of the victim agent. To learn adversarial policy with BIA, we employed DAC (Kostrikov et al., 2018) of ILfD and OPOLO (Zhu et al., 2020) for ILfO as imitation learning algorithms, which are variants of GAIL, and known to be more stable. We used 20 episodes as demonstrations.

Table 1: Comparison of attack performances. Each value is the average episode rewards $\pm$ the standard deviation over 50 episodes. The Target Reward is the reward obtained by the target policy. Attack Rewards are the rewards obtained by the victim under the attack. The attack budget was set to $\epsilon = 0.3$.

| Adv Task | Target Reward | Attack Rewards ($\uparrow$) | | | | |
|---|---|---|---|---|---|---|
| | | Random | Targeted PGD | Reward Maximization | BIA-ILfD (ours) | BIA-ILfO (ours) |
| window-close | $4543 \pm 39$ | $947 \pm 529$ | $1057 \pm 652$ | $\mathbf{4505 \pm 65}$ | $3962 \pm 666$ | $4036 \pm 510$ |
| window-open | $4508 \pm 121$ | $322 \pm 261$ | $296 \pm 117$ | $506 \pm 444$ | $\mathbf{566 \pm 523}$ | $557 \pm 679$ |
| drawer-close | $4868 \pm 6$ | $1069 \pm 1585$ | $1031 \pm 1437$ | $4658 \pm 747$ | $\mathbf{4760 \pm 640}$ | $4626 \pm 791$ |
| drawer-open | $4713 \pm 16$ | $841 \pm 357$ | $825 \pm 326$ | $1499 \pm 536$ | $\mathbf{1556 \pm 607}$ | $1445 \pm 610$ |

**Attack Baselines** We compare our proposed attack method with three baselines. **(i) Random Attack:** This attack perturbs the victim's state observation by random noise drawn from a uniform distribution. **(ii) Targeted PGD Attack:** This naive attack method optimizes fake states $\hat{s}$ using PGD (Projected Gradient Descent) to align the victim's actions with those of the target policy's at each time: $\hat{s} = \arg\min_{\hat{s}} d(\pi(\cdot|\hat{s}), \pi_{\text{tgt}}(\cdot|s))$. *It is important to note that PGD requires white-box access to the victim's policy, giving the adversary an advantage not available in our proposed method.* **(iii) Target Reward Maximization Attack:** This attack leverages Lemma 1 to learn an adversarial policy that maximizes the victim's cumulative reward from the reward function $R_{\text{tgt}}$ which is used when learning the target policy: $\nu^* = \arg\max_{\pi \circ \nu}^M [R_{\text{tgt}}(s, a, s')]$. This attack requires access to the reward function used during the target policy's learning, which is not allowed in our proposed attack. *Thus, this attack also gives the adversary an advantage not available in our proposed method.*

**Defense Baselines** No defense methods have yet been proposed specifically for behavior-targeted attacks. Therefore, we use three defense methods designed for untargeted attacks as baselines: **(i) SA-PPO** (Zhang et al., 2020b): This method aims to increase the smoothness of the policy's action outputs by a regularizer. *The difference between TDRT-PPO and SA-PPO is that SA-PPO does not apply time discounting in the regularization.* **(ii) ATLA-PPO** (Zhang et al., 2021): This is an adversarial training method that alternately trains the adversary and the victim. **(iii) WocaR-PPO** (Liang et al., 2022): This method combines adversarial training and smooth regularization.

## 7.2 ATTACK PERFORMANCE RESULTS

We provide the results in Table 1. Target reward represents the cumulative reward obtained by the target policy in the adversary's task, serving as the upper bound for attack rewards. Our BIA in the black-box model shows superior performance compared to Targeted PGD in the white-box model. Moreover, BIA shows performance that is nearly equivalent to Target Reward Maximization, which directly uses the reward functions when learning the target policy. The performance of BIA with ILfO (i.e., no-box) is lower than that with ILfD (i.e., black-box) because ILfO is not allowed to access the victim's actions.

Interestingly, Targeted PGD showed almost identical performance to Random Attack. Our investigation revealed that, even after optimization by PGD, the loss remains high at several points in the trajectory. As mentioned in Section 5, the attack is likely to fail because no false states that can perfectly align with the actions specified by the target policy exist, especially when epsilon is small. All attack methods show relatively low attack performance in the window-open and drawer-open tasks. This can be attributed to the greater disparity in the state-action distributions between victim policy and the target policy in these tasks compared to other tasks.

## 7.3 DEFENSE PERFORMANCE RESULTS

The results are shown in Table 2. Clean reward represents the cumulative reward obtained by the victim in their original task without attacks; higher values indicate less performance degradation due to robust learning. Best Attack Rewards represent the largest cumulative reward among the five attacks listed in Table 1 when the adversary attempts to force the victim to attain the adversary's target task; lower values indicate greater robustness. Complete results for all attack methods are shown in Table 3.

Table 2: Comparison of Defense Methods. Each value is the average episode rewards $\pm$ the standard deviation over 50 episodes. Clean Rewards are the rewards for the victim's tasks (no attack). Best Attack Rewards shows the highest reward among the five types of adversarial attacks. The attack budget was set to $\epsilon = 0.3$.

| Adv Task | Methods | Clean Rewards ($\uparrow$) | Best Attack Rewards ($\downarrow$) |
|---|---|---|---|
| | PPO | $4508 \pm 121$ | $4505 \pm 65$ |
| window-close | ATLA-PPO (Zhang et al., 2021) | $4169 \pm 467$ | $4270 \pm 188$ |
| | WocaR-PPO (Liang et al., 2022) | $2879 \pm 1256$ | $575 \pm 135$ |
| | SA-PPO (Zhang et al., 2020b) | $4367 \pm 107$ | $485 \pm 61$ |
| | TDRT-PPO (ours) | $4412 \pm 55$ | $\mathbf{482 \pm 3}$ |
| | PPO | $4543 \pm 39$ | $566 \pm 523$ |
| window-open | ATLA-PPO (Zhang et al., 2021) | $4566 \pm 80$ | $586 \pm 649$ |
| | WocaR-PPO (Liang et al., 2022) | $3645 \pm 1575$ | $295 \pm 18$ |
| | SA-PPO (Zhang et al., 2020b) | $4092 \pm 461$ | $272 \pm 37$ |
| | TDRT-PPO (ours) | $4383 \pm 57$ | $\mathbf{254 \pm 214}$ |
| | PPO | $4714 \pm 16$ | $4760 \pm 640$ |
| drawer-close | ATLA-PPO (Zhang et al., 2021) | $4543 \pm 102$ | $4858 \pm 6$ |
| | WocaR-PPO (Liang et al., 2022) | $4193 \pm 304$ | $4867 \pm 8$ |
| | SA-PPO (Zhang et al., 2020b) | $2156 \pm 453$ | $\mathbf{4 \pm 2}$ |
| | TDRT-PPO (ours) | $4237 \pm 93$ | $4860 \pm 4$ |
| | PPO | $4868 \pm 6$ | $1556 \pm 607$ |
| drawer-open | ATLA-PPO (Zhang et al., 2021) | $4863 \pm 7$ | $1158 \pm 1026$ |
| | WocaR-PPO (Liang et al., 2022) | $4704 \pm 654$ | $579 \pm 15$ |
| | SA-PPO (Zhang et al., 2020b) | $4161 \pm 1537$ | $403 \pm 49$ |
| | TDRT-PPO (ours) | $4802 \pm 27$ | $\mathbf{378 \pm 10}$ |

TDRT-PPO shows the highest robustness in three out of the four tasks. Both SA-PPO and TDRT-PPO were tested using the same regularization coefficients. Overall, we observed no significant difference in robustness between the two methods, while TDRT-PPO consistently achieved higher Clean Rewards. These results demonstrate two advantages of time-discounted regularization. First, using TDR does not compromise robustness against attacks, as reflected in Best Attack Rewards. Second, TDR mitigates performance degradation, as seen in Clean Rewards. In the following sections, we evaluate the impact of TDR on performance in more detail.

Overall, methods that regularize the output of the policy with respect to input changes (i.e., SA-PPO, TDRT-PPO, WocaR-PPO) show better robustness, while adversarial training-based defense (i.e., ATLA-PPO) lacks robustness against behavior-targeted attacks. This result is likely because adversarial perturbations used in ALTA-PPO are designed for untargeted attacks, making them less effective than other types of attacks. In the drawer-close task, only SA-PPO demonstrates strong robustness. This suggests that the later stages of the trajectory are critical for attacks in this task, and time discounting may have a negative impact on defense in this task. On the other hand, the Clean Reward of SA-PPO for this task is significantly lower than others, suggesting SA-PPO struggles to maintain a good balance between Clean Rewards and Best Attack Rewards in this task. For further analysis, we present experiments with multiple regularization coefficients in Appendix G.4.

## 8 CONCLUSION

In this work, we proposed the Behavior Imitation Attack (BIA), which manipulates victim behavior through perturbed state observations using imitation learning and operates with limited victim policy access. We also introduced Time-Discounted Regularization (TDR), the first defense method specifically designed for behavior-targeted attacks, which achieved robustness without compromising normal performance.

**Limitations and Future work.** Behavior-targeted attacks cannot force victims to perform behaviors that they would never execute under any observation. When the target policy's behavior significantly deviates from the victim's original patterns, appropriate adversarial states may not exist, leading to attack failure. Additionally, while TDR is effective against behavior-targeted attacks, it shows limited effectiveness against untargeted attacks in the later stages of the trajectory, making the development of defense methods effective against both types of attacks an important future research direction.

ETHICS STATEMENT

This paper focused on adversarial attacks on reinforcement learning and their defense methods, aiming to improve the reliability of deep reinforcement learning. Our contribution lies in proposing attack methods and defenses that are envisioned for real-world scenarios. However, our proposed methods still face challenges in practical real-world applications. Our research fully complies with legal and ethical standards, and there are no conflicts of interest. Throughout this study, we utilized only publicly available benchmarks. No private datasets were used in this research. For reproducibility, we have made our experimental code public.

REPRODUCIBILITY STATEMENT

We provide the complete experimental code that allows for the full reproduction of the experiments presented in this paper: link. Additionally, we have provided detailed algorithms for our proposed methods. The benchmarks used in our experiments are public and accessible to everyone. As described above, we have made substantial efforts to ensure the reproducibility of our results.

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

# A   RELATED WORKS

In this section, we discuss prior research on attacks and defenses on DRL that mainly assume an adversary's perturbed the victim's state observations.

## A.1   UNTARGETED ATTACK AND ITS DEFENSE

In untargeted attacks, the adversary's goal is to minimize the cumulative reward received by the victim. The most basic approach is gradient-based methods. Huang et al. (2017) proposed an attack method that uses the Fast Gradient Signed Method (FGSM) (Goodfellow et al., 2014) to compute adversarial perturbations that prevent the victim from choosing optimal actions. Pattanaik et al. (2018) performed a more powerful attack by creating adversarial perturbations that minimize the value estimated by the Q-function. (Gleave et al., 2020) proposed an attack that assumes a two-agent environment. This method creates an adversarial agent that maximally degrades the victim's performance.

One approach to learning robust policies against untargeted attacks is to make the policy smooth. Shen et al. (2020) introduced a regularization term to smooth the policy and demonstrated increased sample efficiency and robustness. Zhang et al. (2020b) formulated the State-Adversarial Markov Decision Process (SA-MDP) to represent situations where an adversary interferes with the victim's state observations. Based on SA-MDP, they showed that regularization to smooth the policy is effective in resisting untargeted attacks.

Another approach for defense is adversarial training. Zhang et al. (2021) showed that the optimal adversarial policy can be learned as a policy in MDP and proposed ATLA, an adversarial training framework that exploits this insight. Sun et al. (2022) proposed a theoretically optimal attack method that finds the optimal direction of perturbation and proposed PA-ATLA as an extension of ATLA, which is efficient even in large state spaces. Oikarinen et al. (2021) proposed a framework for training a robust RL by incorporating an adversarial loss that accounts for the worst-case input perturbations. Additionally, they introduced a new metric to efficiently evaluate the robustness of the agent. Liang et al. (2022) efficiently estimated the lower bound of cumulative rewards under adversarial attacks and performed adversarial learning with partial smoothness regularization. However, Korkmaz (2021; 2023) pointed out that agents trained to be adversarially robust are still vulnerable to attacks that were not anticipated during training.

McMahan et al. (2024) proposed a comprehensive framework for computing optimal attacks and defenses, modeling the attack problem as a meta-MDP and the defense problem as a partially observable turn-based stochastic game. Bukharin et al. (2024) proposed a robust Multi-Agent RL (MARL) framework that uses adversarial regularization to promote Lipschitz continuity of policies, thereby enhancing robustness against environmental changes, observation noise, and malicious agent actions. Liang et al. (2024) introduced a new concept called temporally-coupled perturbations, where consecutive perturbations are constrained. Their proposed method, GRAD, demonstrates strong robustness against standard and temporally-coupled perturbations. Belaire et al. (2024) proposed a novel defense method based on regret minimization. Liu et al. (2024) have proposed a new framework called "PROTECTED." This framework is characterized by its effectiveness against not only worst-case attacks but also more general forms of attacks. Additionally, another approach in robust reinforcement learning is certified defense Wu et al. (2022); Kumar et al. (2022); Mu et al. (2024); Sun et al. (2024). These studies guarantee a lower bound on the rewards obtained by the victim under adversarial attacks.

## A.2   TARGETED ATTACK

The adversary's goal of the targeted attack is to manipulate the victim so that it follows some direction made by the adversary. One type of targeted attack is the enchanting attack. This attack aims to lure the victim agent to reach a predetermined target state. Lin et al. (2017) first proposed this type of attack. In their approach, the adversary predicts future states and generates a sequence of actions that cause the victim to reach the target state. The adversary then crafts a sequence of perturbations to make the victim perform the sequence of actions. Tretschk et al. (2018) proposed an enchanting attack where adversarial perturbations to maximize adversarial rewards are heuristically designed for the attack purpose, thereby transitioning to the specified target state. Buddareddygari

et al. (2022) proposed a different enchanting attack using visual patterns placed on physical objects in the environment so that the victim agent is directed to the target state. Unlike (Lin et al., 2017) and (Tretschk et al., 2018), which affect the victim's state observations, this attack affects the environmental dynamics, not the victim's state observations. Ying et al. (2023) proposed an enchanting attack with a universal adversarial perturbation. When any state observations are modified with this perturbation, the victim agent is forced to be guided to the target state. All of these enchanting attacks require white-box access to the victim's policy.

The other type of targeted attack is the behavior-targeted attack, which aims to manipulate not only the final destination but also the behavior in a more detailed direction by the adversary. Hussenot et al. (2020) proposed a behavior-targeted method that forces the victim to select the same actions as the policy specified by the adversary. More specifically, this attack precomputes a universal perturbation for each action so that the victim who observes the perturbed states takes the same action as the adversarially specified policy. One limitation of this attack is that the computation cost to precompute such universal perturbations can be large when the action space is large or continuous. Since the cost required for pre-computing perturbations is significant, applying this attack is challenging. Boloor et al. (2020) proposed a heuristic attack specifically designed for autonomous vehicles. They formulated an objective function with a detailed knowledge of the target task, which is not applicable to behavior-targeted attacks for general tasks. We remark that all existing behavior-targeted attacks also require white-box access to the victim's policy.

### A.3 POISONING ATTACK

Several studies have proposed poisoning attacks against RL agents, which intervene during the victim's training phase. Sun et al. (2021) proposed VA2C-P, a poisoning attack framework that can adapt without access to environment dynamics. They proposed both untargeted attacks and target attacks that aim to align the victim's policy with a target policy.Rakhsha et al. (2020) theoretically investigated the attack problem in both offline planning and online RL settings with tabular MDPs. Rangi et al. (2022) also provided theoretical insights into the fundamental limits of poisoning attacks in episodic reinforcement learning. Xu et al. (2023); Xu & Singh (2023) proposed more practical black-box attacks that do not require knowledge of environment dynamics or the victim's learning algorithm. As our research focuses on attacking trained victims, our threat model differs from those of these studies.

## B THEORITICAL ANALYSIS OF THE DISTRIBUTION MATCHING APPROACH IN OUR ADVERSARIAL ATTACK SETTING

In this section, we prove that the distribution matching approach in our problem setting is equivalent to the problem of minimizing the distance between policies through inverse reinforcement learning. This shows that the adversarial policy, which is learned through our distribution matching approach, rigorously mimics the target policy. Our procedures follow the proof of Proposition 3.1 in (Ho & Ermon, 2016).

Let a set of all stationary stochastic policies as $\Pi$ and a set of all stationary stochastic adversarial policies as $\mathcal{N}$. Also, we write $\bar{\mathbb{R}}$ for extended real numbers $\mathbb{R} \cup \{\infty\}$. The goal of inverse reinforcement learning is to find a reward function such that when a policy is learned to maximize the rewards obtained from this function, it matches the expert's policy. This process aims to derive a reward function from the expert's trajectories. We formulate the adversary's objective as learning an adversarial policy that maximizes the victim's cumulative reward from the reward function estimated by IRL:

$$\mathrm{IRL}_{\psi,\nu}(\pi_{\mathrm{tgt}}) = \arg\max_{c \in \mathbb{R}^{\mathcal{S} \times \mathcal{A}}} -\psi(c) + \left( \min_{\nu \in \mathcal{N}} \mathbb{E}^M_{\pi \circ \nu}[c(s,a)] \right) - \mathbb{E}^M_{\pi_{\mathrm{tgt}}}[c(s,a)], \qquad (15)$$

$$\mathrm{RL}(c) = \arg\min_{\nu \in \mathcal{N}} \mathbb{E}_{\pi \circ \nu}[c(s,a)], \qquad (16)$$

where $c : \mathcal{S} \times \mathcal{A} \to \mathbb{R}$ is a cost function, $\psi : \mathbb{R}^{\mathcal{S} \times \mathcal{A}} \to \bar{\mathbb{R}}$ is a convex cost function regularization. Note that we use a cost function instead of a reward function to represent reinforcement learning

as a minimization problem. Let $c^* \in \text{IRL}_{\psi,\nu}(\pi_{\text{tgt}})$ be the optimal cost function through IRL. The optimal adversarial policy is learned with respect to the optimal cost function: $\nu^* \in \text{RL}(c^*)$.

For the proof, we define the occupancy measure. The occupancy measure is an unnormalized state-action distribution:

$$\hat{\rho}_\pi(s,a) = \sum_{t=0}^{\infty} \gamma^t \Pr\left(s_t = s, a_t = a | s_0 \sim p_0(\cdot), a_t \sim \pi(\cdot|s_t), s_{t+1} \sim p(\cdot|s_t, a_t)\right). \tag{17}$$

Therefore, we present the Theorem 2. The Theorem shows that the optimal adversarial policy obtained via inverse reinforcement learning coincides with the optimal adversarial policy obtained through the distribution matching approach:

**Theorem 2.** *Let $\mathcal{C}$ be the set of cost functions, $\psi$ be a convex function and $\psi^*$ be the conjugate of $\psi$. When $\pi$ is fixed and $\mathcal{C}$ is a compact convex set, the following holds:*

$$\text{RL} \circ \text{IRL}_{\psi,\nu}(\pi_{tgt}) = \underset{\nu \in \mathcal{N}}{\arg\min}\, \psi^*(\hat{\rho}_{\pi \circ \nu} - \hat{\rho}_{\pi_{tgt}}). \tag{18}$$

*Proof.* Let $\mathcal{D}_{\pi \circ \nu} = \{\hat{\rho}_{\pi \circ \nu} | \nu \in \mathcal{N}\}$. If $\mathcal{D}_{\pi \circ \nu}$ is a compact and convex set, equation 18 is valid according to the proof of Theorem 3.1 in (Ho & Ermon, 2016). Thus, we prove that $\mathcal{D}_{\pi \circ \nu}$ is a compact and convex set.

**Compactness:** The mapping from $\nu$ to $\pi \circ \nu$ is linear, and $\mathcal{N}$ is compact. Therefore, by (Arkhangel'skiĭ & Fedorchuk, 1990), $\Pi_\nu$ is a compact set. From (Ho & Ermon, 2016), when $\Pi_\nu$ is compact, $\mathcal{D}_{\pi \circ \nu}$ is also compact. Consequently, $\mathcal{D}_{\pi \circ \nu}$ is a compact set.

Next, we show that $\mathcal{D}_{\pi \circ \nu}$ is closed. Policy $\pi \in \Pi$ and occupancy measure $\hat{\rho} \in \mathcal{D}$ have a one-to-one correspondence by Lemma 1 in (Ho & Ermon, 2016). Let $\Pi_\nu = \{\pi' \mid \nu \in \mathcal{N}, \pi' = \pi \circ \nu\}$ be the set of all behavior policies. Since $\Pi_\nu \subseteq \Pi$, $\pi \circ \nu \in \Pi_\nu$ and $\hat{\rho}_{\pi \circ \nu} \in \mathcal{D}_{\pi \circ \nu}$ also have a one-to-one correspondence. Thus, let $\{\hat{\rho}_{\pi \circ \nu_1}, \hat{\rho}_{\pi \circ \nu_2}, \dots\}$ be any cauchy sequence with $\hat{\rho}_{\pi \circ \nu_n} \in \mathcal{D}_{\pi \circ \nu}$. Due to the one-to-one correspondence, there exists a corresponding sequence $\{\pi \circ \nu_1, \pi \circ \nu_2, \dots\}$. Since $\Pi_\nu$ is compact, the sequence $\{\pi \circ \nu_1, \pi \circ \nu_2, \dots\}$ converges to $\pi \circ \nu \in \Pi_\nu$. Hence, the sequence $\{\hat{\rho}_{\pi \circ \nu_1}, \hat{\rho}_{\pi \circ \nu_2}, \dots\}$ also converges to the occupancy measure $\hat{\rho}_{\pi \circ \nu}$ corresponding to $\pi \circ \nu$. Therefore, $\mathcal{D}_{\pi \circ \nu}$ is closed.

**Convexity:** First, we show that $\Pi_\nu$ is a convex set. For $\forall \nu_1 \in \mathcal{N}, \forall \nu_2 \in \mathcal{N}$ and $\lambda \in [0,1]$, we define $\pi \circ \nu$ as

$$\pi \circ \nu = \lambda \pi \circ \nu_1 + (1-\lambda)\pi \circ \nu_2. \tag{19}$$

Then, we have

$$\pi \circ \nu = \lambda \pi \circ \nu_1 + (1-\lambda)\pi \circ \nu_2 \tag{20}$$

$$= t \sum_{\hat{a} \in \mathcal{S}} \nu_1(\hat{a}|s)\pi(a|\hat{a}) + (1-t)\sum_{\hat{a} \in \mathcal{S}} \nu_2(\hat{a}|s)\pi(a|\hat{a}) \tag{21}$$

$$= \sum_{\hat{a} \in \mathcal{S}} (t\nu_1(\hat{a}|s) + (1-t)\nu_2(\hat{a}|s))\pi(a|\hat{a}) \tag{22}$$

$$= \pi \circ (\lambda \nu_1 + (1-\lambda)\nu_2) \tag{23}$$

Using the convexity of $\mathcal{N}$, we have $t\nu_1 + (1-t)\nu_2 \in \mathcal{N}$. Thus, $\pi \circ \nu \in \Pi_\nu$ holds for any $\nu_1 \in \mathcal{N}, \nu_2 \in \mathcal{N}$, and $\lambda \in [0,1]$ and $\Pi_\nu$ is convex.

Noting that the one-to-one correspondence of $\hat{\rho}_{\pi \circ \nu} \in \mathcal{D}_{\pi \circ \nu}$ and $\pi \circ \nu \in \Pi_\nu$, for any mixture policy $\pi \circ \nu_{\text{m}} \in \Pi_\nu$, we have $\hat{\rho}_{\pi \circ \nu_{\text{m}}} \in \mathcal{D}_{\pi \circ \nu}$. Consequently, $\mathcal{D}_{\pi \circ \nu}$ is a convex set.

Based on the above, we prove the Theorem following the same procedure as the proof of Theorem 3.1 in (Ho & Ermon, 2016). Let $\tilde{c} \in \text{IRL}_{\psi,\nu}(\pi_{\text{tgt}}), \widetilde{\pi \circ \nu} \in \text{RL}(\tilde{c}) = \text{RL} \circ \text{IRL}_{\psi,\nu}(\pi_{\text{tgt}})$. The RHS of 18 is denoted by

$$\pi \circ \nu_A \in \underset{\pi \circ \nu}{\arg\min}\, \psi^*(\hat{\rho}_{\pi \circ \nu} - \hat{\rho}_{\pi_{\text{tgt}}}) = \underset{\pi \circ \nu}{\arg\min} \max_c -\psi(c) + \int_{s,a} (\hat{\rho}_{\pi \circ \nu}(s,a) - \hat{\rho}_{\pi_{\text{tgt}}}(s,a))c(s,a).$$

$$\tag{24}$$

We define the RHS of 18 by $\bar{L} : \mathcal{D}_{\pi \circ \nu} \times \mathcal{C} \to \mathbb{R}$ as follow:

$$\bar{L}(\hat{\rho}, c) = -\psi(c) + \int_{s,a} \hat{\rho}(s,a) c(s,a) - \int_{s,a} \hat{\rho}_{\pi_{\text{adv}}} c(s,a). \tag{25}$$

We remark that $\bar{L}$ takes an occupancy measure as its argument.

To prove $\widetilde{\pi \circ \nu} = \pi \circ \nu_A$, we utilize the minimax duality of $\bar{L}$.

Policy $\pi \in \Pi$ and occupancy measure $\hat{\rho} \in \mathcal{D}$ have a one-to-one correspondence by Lemma 1 in (Ho & Ermon, 2016). Since $\Pi_\nu \subseteq \Pi$, $\pi \circ \nu \in \Pi_\nu$ and $\hat{\rho}_{\pi \circ \nu} \in \mathcal{D}_{\pi \circ \nu}$ also have a one-to-one correspondence. Thus, the following relationship is established:

$$\hat{\rho}_{\pi \circ \nu_A} \in \arg\min_{\hat{\rho} \in \mathcal{D}_{\pi \circ \nu}} \max_{c \in \mathcal{C}} \bar{L}(\hat{\rho}, c), \tag{26}$$

$$\tilde{c} \in \arg\max_{c \in \mathcal{C}} \min_{\hat{\rho} \in \mathcal{D}_{\pi \circ \nu}} \bar{L}(\hat{\rho}, c), \tag{27}$$

$$\hat{\rho}_{\widetilde{\pi \circ \nu}} \in \arg\min_{\hat{\rho} \in \mathcal{D}_{\pi \circ \nu}} \bar{L}(\hat{\rho}, \tilde{c}). \tag{28}$$

$\mathcal{D}_{\pi \circ \nu}$ is a compact convex set and $\mathcal{C}$ is also a compact convex set. Since $\psi$ is a convex function, we have that $\bar{L}(\cdot, c)$ is convex for all $c$, and that $\bar{L}(\hat{\rho}, \cdot)$ is liner for all $\hat{\rho}$, so $\bar{L}(\hat{\rho}, \cdot)$ is concave for all $\hat{\rho}$. Due to minimax duality(Fernique et al., 1983), we have the following equality:

$$\min_{\hat{\rho} \in \mathcal{D}_{\pi \circ \nu}} \max_{c \in \mathcal{C}} \bar{L}(\hat{\rho}, c) = \max_{c \in \mathcal{C}} \min_{\hat{\rho} \in \mathcal{D}_{\pi \circ \nu}} \bar{L}(\hat{\rho}, c). \tag{29}$$

Therefore, from equation 26 and equation 27, $(\hat{\rho}_{\pi \circ \nu_A}, \tilde{c})$ is a saddle point of $\bar{L}$, which implies that $\hat{\rho}_{\pi \circ \nu_A} \in \arg\min_{\hat{\rho} \in \mathcal{D}_{\pi \circ \nu}} \bar{L}(\hat{\rho}, \tilde{c})$ and so $\hat{\rho}_{\widetilde{\pi \circ \nu}} = \hat{\rho}_{\pi \circ \nu_A}$. $\qquad \square$

The right-hand side of equation 18 represents the optimal adversarial policy that minimizes the distance between occupancy measures as measured by $\psi^*$. Therefore, Theorem 2 indicates that the optimal adversarial policy obtained through the distribution matching approach is equivalent to that via inverse reinforcement learning.

## C PROOFS

### C.1 PROOF OF LEMMA 1

**Lemma 1.** *Consider an SA-MDP $M = (\mathcal{S}, \mathcal{A}, R, \mathcal{B}, p, \gamma)$, and let $\pi$ be the victim's policy. Given the reward function $\bar{R}$ specified by the adversary, the reward function $\hat{R}$ and state transition probability $\hat{p}$ are defined as follows:*

$$\hat{R}(s, \hat{s}, s') = \mathbb{E}[\hat{r}|s, \hat{s}, s'] = \begin{cases} \frac{\sum_{a \in \mathcal{A}} \pi(a|\hat{s}) p(s'|s,a) \bar{R}(s,a,s')}{\sum_{a \in \mathcal{A}} \pi(a|\hat{s}) p(s'|s,a)} & \text{if } \hat{s} \in \mathcal{B}(s) \\ C & \text{otherwise,} \end{cases} \tag{9}$$

$$\hat{p}(s'|s, \hat{s}) = \sum_{a \in \mathcal{A}} \pi(a|\hat{s}) p(s'|s, a). \tag{10}$$

*Then, for the MDP $\hat{M} = (\mathcal{S}, \mathcal{S}, \hat{R}, \hat{p}, \gamma)$, the following equality holds:*

$$\nu^* = \arg\max_\nu \mathbb{E}_\nu^{\hat{M}}[\hat{R}(s, \hat{s}, s')] = \arg\max_\nu \mathbb{E}_{\pi \circ \nu}^M[\bar{R}(s, a, s')]. \tag{11}$$

*Proof.* This proof follows the approach outlined in the proof of Lemma 1 in (Zhang et al., 2020b), with some modifications to account for the differences in our setting.

In the proof of Lemma 1 presented in [1], by substituting $-R(s, a, s')$ with $\bar{R}(s, a, s')$, we can derive the subsequent results:

$$\hat{R}(s, \hat{a}, s') = \frac{\sum_a \bar{R}(s, a, s') p(s'|a, s) \pi(a|\hat{a})}{\sum_a p(s'|a, s) \pi(a|\hat{a})}. \tag{30}$$

Let $\overline{M} = \max_{s,a,s'} \bar{R}(s,a,s')$ and $\underline{M} = \min_{s,a,s'} \bar{R}(s,a,s')$. We Define the reward $C$ for when the adversarial policy selects an action $\hat{a} \notin \mathcal{B}$ as follows:

$$C < \min \left\{ \underline{M}, \frac{1}{1-\gamma}\underline{M} - \frac{\gamma}{1-\gamma}\overline{M} \right\}. \tag{31}$$

From the definition of $C$ and $\overline{M}$, we have for $\forall(s, \hat{a}, s')$,

$$C < \hat{R}(s, \hat{a}, s') \leq \overline{M}, \tag{32}$$

and, for $\forall \hat{a} \in \mathcal{B}(s)$, according to equation 30,

$$\underline{M} \leq \hat{R}(s, \hat{a}, s') \leq \overline{M}. \tag{33}$$

MDP has at least one optimal policy, so the $\hat{M}$ has an optimal adversarial policy $\nu^*$, which satisfies $\hat{V}_{\pi \circ \nu^*}(s) \geq \hat{V}_{\pi \circ \nu}(s)$ for $\forall s, \forall \nu$. From the property of the optimal policy, $\nu^*$ is deterministic. Let $\mathfrak{R} \triangleq \{\nu \mid \forall s, \exists \hat{a} \in \mathcal{B}(s), \nu(\hat{a} \mid s) = 1\}$. This restricts that the adversarial policy does not take actions $\hat{a} \notin B(s)$, so $\nu^* \in \mathfrak{R}$. If $\nu^* \notin \mathfrak{R}$ at state $s^0$,

$$\hat{V}_{\pi \circ \nu^*}(s^0) = \mathbb{E}_{\hat{p},\nu^*} \left[ \sum_{k=0}^{\infty} \gamma^k \hat{r}_{t+k+1} \mid s_t = s^0 \right] \tag{34}$$

$$= C + \mathbb{E}_{\hat{p},\nu^*} \left[ \sum_{k=1}^{\infty} \gamma^k \hat{r}_{t+k+1} \mid s_t = s^0 \right] \tag{35}$$

$$\leq C + \frac{\gamma}{1-\gamma}\overline{M} \tag{36}$$

$$< \frac{1}{1-\gamma}\overline{M} \tag{37}$$

$$\leq \mathbb{E}_{\hat{p},\nu'} \left[ \sum_{k=0}^{\infty} \gamma^k \hat{r}_{t+k+1} \mid s_t = s^0 \right] = \hat{V}_{\pi \circ \nu'}(s^0). \tag{38}$$

The last inequality holds for any $\nu' \in \mathfrak{R}$. This contradicts the assumption that $\nu^*$ is optimal. Hence, the following analysis will only consider policies included in $\mathcal{N}$.

For any policy $\nu \in \mathfrak{R}$:

$$\hat{V}_{\pi \circ \nu}(s) = \mathbb{E}_{\hat{p},\nu} \left[ \sum_{k=0}^{\infty} \gamma^k \hat{r}_{t+k+1} \mid s_t = s \right] \tag{39}$$

$$= \mathbb{E}_{\hat{p},\nu} \left[ \hat{r}_{t+1} + \gamma \sum_{k=0}^{\infty} \gamma^k \hat{r}_{t+k+2} \mid s_t = s \right] \tag{40}$$

$$= \sum_{\hat{a} \in \mathcal{S}} \nu(\hat{a}|s) \sum_{s' \in \mathcal{S}} \hat{p}(s'|s,\hat{a}) \left[ \hat{R}(s,\hat{a},s') + \gamma \mathbb{E}_{\hat{p},\nu} \left[ \sum_{k=0}^{\infty} \gamma^k \hat{r}_{t+k+2} \mid s_{t+1} = s' \right] \right] \tag{41}$$

$$= \sum_{\hat{a} \in \mathcal{S}} \nu(\hat{a}|s) \sum_{s' \in \mathcal{S}} \hat{p}(s'|s,\hat{a}) \left[ \hat{R}(s,\hat{a},s') + \gamma \hat{V}_{\pi \circ \nu}(s') \right]. \tag{42}$$

All policies in $\mathfrak{R}$ are deterministic, so we denote the deterministic action $\hat{a}$ chosen by a $\nu \in \mathfrak{R}$ at $s$ as $\nu(s)$. Then for $\forall \nu \in \mathfrak{R}$, we have

$$\hat{V}_{\pi \circ \nu}(s) = \sum_{s' \in \mathcal{S}} \hat{p}(s'|s,\hat{a}) \left[ \hat{R}(s,\hat{a},s') + \gamma \hat{V}_{\pi \circ \nu}(s') \right] \tag{43}$$

$$= \sum_{s' \in \mathcal{S}} \sum_{a \in \mathcal{A}} \pi(a|\hat{a}) p(s'|s,a) \left[ \frac{\sum_a \bar{R}(s,a,s') p(s'|a,s) \pi(a|\hat{a})}{\sum_a p(s'|a,s) \pi(a|\hat{a})} + \gamma \hat{V}_{\pi \circ \nu}(s') \right] \tag{44}$$

$$= \sum_{a \in \mathcal{A}} \pi(a|\nu(s)) \sum_{s' \in \mathcal{S}} p(s'|s,a) \left[ \bar{R}(s,a,s') + \gamma \hat{V}_{\pi \circ \nu}(s') \right]. \tag{45}$$

Thus, the optimal value function is

$$\hat{V}_{\pi \circ \nu^*}(s) = \max_{\nu^*(s) \in \mathcal{B}(s)} \sum_{a \in \mathcal{A}} \pi(a|\nu^*(s)) \sum_{s' \in \mathcal{S}} p(s'|s, a) \left[ \bar{R}(s, a, s') + \gamma \hat{V}_{\pi \circ \nu^*}(s') \right]. \quad (46)$$

The Bellman equation for the state value function $\tilde{V}_{\pi \circ \nu}(s)$ of the SA-MDP $M = (\mathcal{S}, \mathcal{A}, \mathcal{B}, R, p, \gamma)$ is given as follows:

**Lemma 2** (Theorem 1 of (Zhang et al., 2020b)). *Given $\pi : \mathcal{S} \to \mathcal{P}(\mathcal{A})$ and $\nu : \mathcal{S} \to \mathcal{S}$, we have*

$$\tilde{V}_{\pi \circ \nu}(s) = \sum_{a \in \mathcal{A}} \pi(a|\nu(s)) \sum_{s' \in \mathcal{S}} p(s'|s, a) \left[ R(s, a, s') + \gamma \tilde{V}_{\pi \circ \nu}(s') \right] \quad (47)$$

Therefore, if the reward function is $\bar{R}$, then $\hat{V}_{\pi \circ \nu^*} = \tilde{V}_{\pi \circ \nu^*}$. So, we have

$$\tilde{V}_{\pi \circ \nu^*}(s) = \max_{\nu^*(s) \in \mathcal{B}(s)} \sum_{a \in \mathcal{A}} \pi(a|\nu^*(s)) \sum_{s' \in \mathcal{S}} p(s'|s, a) \left[ \bar{R}(s, a, s') + \gamma \tilde{V}_{\pi \circ \nu^*}(s') \right], \quad (48)$$

and $\tilde{V}_{\pi \circ \nu^*}(s) \geq \tilde{V}_{\pi \circ \nu}(s)$ for $\forall s, \forall \nu \in \mathfrak{R}$. Hence, $\nu^*$ is also the optimal $\nu$ for $\tilde{V}_{\pi \circ \nu}$.

$\square$

### C.2 PROOF OF THEOREM 1

**Theorem 1.** *Let the discount factor be $\gamma \in (0, 1)$. For state-action distributions $\rho_\pi$ and $\rho_{\pi \circ \nu}$ corresponding to policies $\pi$ and $\pi \circ \nu$, the following inequality holds:*

$$D_{KL}(\rho_\pi || \rho_{\pi \circ \nu}) \leq \frac{1}{1 - \gamma} \sum_{t=0}^{\infty} \gamma^t \mathbb{E}_{s \sim d_\pi^t} [D_{KL}(\pi(\cdot|s) || \pi \circ \nu(\cdot|s))], \quad (13)$$

*where $d_\pi^t(s) = \Pr(s_t = s|\pi)$ represents the state distribution of $\pi$ at time $t$.*

*Proof.* We first introduce the state distribution to prove Theorem 1. The state distribution $d_\pi : \mathcal{S} \to [0, 1]$ represents the probability of encountering a specific state when transitioning according to a policy $\pi$:

$$d_\pi(s) = (1 - \gamma) \sum_{t=0}^{\infty} \gamma^t \Pr \left( s_t = s | s_0 \sim p_0(\cdot), a_t \sim \pi(\cdot|s_t), s_{t+1} \sim p(\cdot|s_t, a_t) \right). \quad (49)$$

We establish the following lemma:

**Lemma 3.** *Given two policies $\pi, \pi \circ \nu : \mathcal{S} \to \mathcal{P}(\mathcal{A})$ and their state distribution $d_\pi, d_{\pi \circ \nu}$, the following inequality holds:*

$$D_{KL}(d_\pi || d_{\pi \circ \nu}) \leq \frac{\gamma^2}{1 - \gamma^2} \sum_{t=1}^{\infty} \gamma^t \mathbb{E}_{s \sim d_\pi^t} [D_{KL}(\pi(\cdot|s) || \pi \circ \nu(\cdot|s))] \quad (50)$$

*Proof.* This proof is based on Theorem 4.1 of (Belkhale et al., 2024). Regarding the distance between the state distributions of $\pi$ and $\pi \circ \nu$ at time $t$ $D_{KL}(d_\pi^t, d_{\pi \circ \nu}^t)$, the following inequality holds

for $t \geq 1$ by using KL's joint convexity and Jensen's inequality:

$$D_{\mathrm{KL}}(d_\pi^t \| d_{\pi \circ \nu}^t) = \int_{s'} d_\pi^t(s') \log \frac{d_\pi^t(s')}{d_{\pi \circ \nu}^t(s')} \tag{51}$$

$$= \int_{s'} \left( \int_{s,a} \gamma d_\pi^{t-1}(s) \pi(a|s) p(s'|s,a) \right) \log \frac{\int_{s,a} \gamma d_\pi^{t-1}(s) \pi(a|s) p(s'|s,a)}{\int_{s,a} \gamma d_{\pi \circ \nu}^{t-1}(s) \pi \circ \nu(a|s) p(s'|s,a)} \tag{52}$$

$$\leq \int_{s'} \int_{s,a} \gamma d_\pi^{t-1}(s) \pi(a|s) p(s'|s,a) \log \frac{\gamma d_\pi^{t-1}(s) \pi(a|s) p(s'|s,a)}{\gamma d_{\pi \circ \nu}^{t-1}(s) \pi \circ \nu(a|s) p(s'|s,a)} \tag{53}$$

$$\leq \int_{s'} \int_{s,a} \gamma d_\pi^{t-1}(s) \pi(a|s) p(s'|s,a) \left( \log \frac{d_\pi^{t-1}(s)}{d_{\pi \circ \nu}^{t-1}(s)} + \log \frac{\pi(a|s)}{\pi \circ \nu(a|s)} \right) \tag{54}$$

$$\leq \gamma \int_{s,a} d_\pi^{t-1}(s) \pi(a|s) \left( \log \frac{d_\pi^{t-1}(s)}{d_{\pi \circ \nu}^{t-1}(s)} + \log \frac{\pi(a|s)}{\pi \circ \nu(a|s)} \right) \tag{55}$$

$$\leq \gamma \int_s d_\pi^{t-1}(s) \log \frac{d_\pi^{t-1}(s)}{d_{\pi \circ \nu}^{t-1}(s)} + \gamma \int_{s,a} d_\pi^{t-1}(s) \pi(a|s) \log \frac{\pi(a|s)}{\pi \circ \nu(a|s)} \tag{56}$$

$$\leq \gamma D_{\mathrm{KL}}(d_\pi^{t-1} \| d_{\pi \circ \nu}^{t-1}) + \gamma \mathbb{E}_{s \sim d_\pi^{t-1}} [D_{\mathrm{KL}}(\pi(\cdot|s) \| \pi \circ \nu(\cdot|s))] \tag{57}$$

$$\leq \gamma^t D_{\mathrm{KL}}(d_\pi^0 \| d_{\pi \circ \nu}^0) + \sum_{j=0}^{t-1} \gamma^{t-j} \mathbb{E}_{s \sim d_\pi^j} [D_{\mathrm{KL}}(\pi(\cdot|s) \| \pi \circ \nu(\cdot|s))] \tag{58}$$

$$\leq \sum_{j=0}^{t-1} \gamma^{t-j} \mathbb{E}_{s \sim d_\pi^j} [D_{\mathrm{KL}}(\pi(\cdot|s) \| \pi \circ \nu(\cdot|s))] \tag{59}$$

Thus, we obtain the following inequality:

$$D_{\mathrm{KL}}(d_\pi \| d_{\pi \circ \nu}) = \int_s (\sum_{t=0}^{\infty} \gamma^t d_\pi^t(s)) \log \frac{\sum_{t=1}^{\infty} \gamma^t d_\pi^t(s)}{\sum_{t=1}^{\infty} \gamma^t d_{\pi \circ \nu}^t(s)} \tag{60}$$

$$\leq \int_s \sum_{t=0}^{\infty} \gamma^t d_\pi^t(s) \log \frac{\gamma^t d_\pi^t(s)}{\gamma^t d_{\pi \circ \nu}^t(s)} \tag{61}$$

$$\leq \sum_{t=0}^{\infty} \gamma^t \int_s d_\pi^t(s) \log \frac{d_\pi^t(s)}{d_{\pi \circ \nu}^t(s)} \tag{62}$$

$$\leq \sum_{t=1}^{\infty} \gamma^t D_{\mathrm{KL}}(d_\pi^t \| d_{\pi \circ \nu}^t) \tag{63}$$

$$\leq \sum_{t=1}^{\infty} \gamma^t \sum_{j=0}^{t-1} \gamma^{t-j} \mathbb{E}_{s \sim d_\pi^j} [D_{\mathrm{KL}}(\pi(\cdot|s) \| \pi \circ \nu(\cdot|s))] \tag{64}$$

$$\leq \sum_{t=1}^{\infty} \sum_{j=0}^{t-1} \gamma^{2t-j} \mathbb{E}_{s \sim d_\pi^j} [D_{\mathrm{KL}}(\pi(\cdot|s) \| \pi \circ \nu(\cdot|s))] \tag{65}$$

$$\leq \frac{\gamma^2}{1-\gamma^2} \sum_{t=1}^{\infty} \gamma^t \mathbb{E}_{s \sim d_\pi^t} [D_{\mathrm{KL}}(\pi(\cdot|s) \| \pi \circ \nu(\cdot|s))] \tag{66}$$

$\square$

Using Lemma 3, the following inequality holds for the distance between state-action distributions:

$$D_{\text{KL}}(\rho_\pi \| \rho_{\pi \circ \nu}) = \int_{s,a} \rho_\pi(s,a) \log \frac{\rho_\pi(s,a)}{\rho_{\pi \circ \nu}(s,a)} \tag{67}$$

$$= \int_{s,a} \pi(a|s) d_\pi(s) \log \frac{\pi(a|s) d_\pi(s)}{\pi \circ \nu(a|s) d_{\pi \circ \nu}(s)} \tag{68}$$

$$= \int_{s,a} \pi(a|s) d_\pi(s) \log \frac{\pi(a|s)}{\pi \circ \nu(a|s)} + \int_{s,a} \pi(a|s) d_\pi(s) \log \frac{d_\pi(s)}{d_{\pi \circ \nu}(s)} \tag{69}$$

$$= \int_{s,a} \pi(a|s) (\sum_{t=0}^\infty \gamma^t d_\pi^t(s)) \log \frac{\pi(a|s)}{\pi \circ \nu(a|s)} + \int_s d_\pi(s) \log \frac{d_\pi(s)}{d_{\pi \circ \nu}(s)} \tag{70}$$

$$= \sum_{t=0}^\infty \gamma^t \int_{s,a} d_\pi^t(s) \pi(a|s) \log \frac{\pi(a|s)}{\pi \circ \nu(a|s)} + D_{\text{KL}}(d_\pi \| d_{\pi \circ \nu}) \tag{71}$$

$$\leq \sum_{t=0}^\infty \gamma^t \mathbb{E}_{s \sim d_\pi^t} [D_{\text{KL}}(\pi(\cdot|s) \| \pi \circ \nu(\cdot|s))] + \frac{\gamma^2}{1-\gamma^2} \sum_{t=1}^\infty \gamma^t \mathbb{E}_{s \sim d_\pi^t} [D_{\text{KL}}(\pi(\cdot|s) \| \pi \circ \nu(\cdot|s))] \tag{72}$$

$$\leq \sum_{t=0}^\infty \gamma^t \mathbb{E}_{s \sim d_\pi^t} [D_{\text{KL}}(\pi(\cdot|s) \| \pi \circ \nu(\cdot|s))] + \frac{\gamma^2}{1-\gamma^2} \sum_{t=0}^\infty \gamma^t \mathbb{E}_{s \sim d_\pi^t} [D_{\text{KL}}(\pi(\cdot|s) \| \pi \circ \nu(\cdot|s))] \tag{73}$$

$$= \frac{1}{1-\gamma^2} \sum_{t=0}^\infty \gamma^t \mathbb{E}_{s \sim d_\pi^t} [D_{\text{KL}}(\pi(\cdot|s) \| \pi \circ \nu(\cdot|s))] \tag{74}$$

$$\square$$

# D  TDRT-PPO ALGORITHM

---

**Algorithm 2** Time-Discouted Robust Training in PPO (TDRT-PPO)

---

**Input:** Number of iterations $T$, clipping parameter $\epsilon_c$, Minibatch size $M$, regularization coefficient $\lambda$
**Output:** Optimized policy $\pi_\theta$
1: Initialize actor network $\pi_\theta(a|s)$ and critic network $V_\phi(s)$
2: **for** $t = 1$ to $T$ **do**
3:     $\mathcal{D} \leftarrow$ Collect trajectories using current policy $\pi_\theta$
4:     **for** each $(s_t, a_t, r_t, s_{t+1})$ in $\mathcal{D}$ **do**
5:         $\hat{R}_t \leftarrow \sum_{l=0}^\infty \gamma^l r_{t+l}$                    ▷ Compute discounted returns
6:         $\hat{A}_t \leftarrow \hat{R}_t - V_\phi(s_t)$                    ▷ Estimate advantages
7:     **end for**
8:     **for** $K$ epochs **do**
9:         $B \leftarrow \{(s_{t_i}, a_{t_i}, \hat{R}_{t_i}, \hat{A}_{t_i})\}_{i=1}^M \sim \mathcal{D}$                    ▷ Sample minibatch of size $M$
10:        $\phi \leftarrow \phi - \alpha_V \nabla_\phi \frac{1}{M} \sum_{i=1}^M (V_\phi(s_i) - \hat{R}_i)^2$                    ▷ Update critic network
11:        # Compute the time-discounted regularization term
12:        $\mathcal{R}_\theta \leftarrow \sum_{s_{t_i} \in B} \max_{\hat{s}_{t_i} \in \mathcal{B}(s_{t_i})} \gamma^{t_i} D_{\text{KL}}(\pi_\theta(\cdot|s_{t_i}) \| \pi_\theta(\cdot|\hat{s}_{t_i}))$
13:        $r_i(\theta) = \frac{\pi_\theta(a_i|s_i)}{\pi_{\theta_{\text{old}}}(a_i|s_i)}$                    ▷ Compute probability ratio
14:        # Update actor-network with regularization
15:        $\theta \leftarrow \theta + \alpha_\pi \nabla_\theta \left( \frac{1}{M} \sum_{i=1}^M \min(r_i(\theta)\hat{A}_i, \text{clip}(r_i(\theta), 1-\epsilon_c, 1+\epsilon_c)\hat{A}_i) + \lambda \mathcal{R}_\theta \right)$
16:     **end for**
17: **end for**
18: **return** Optimized parameters $\theta, \phi$

---

We present the complete algorithm for TDRT-PPO in Algorithm 2. This algorithm extends the standard PPO algorithm by incorporating time-discounted regularization $\mathcal{R}_\theta$. For calculating the

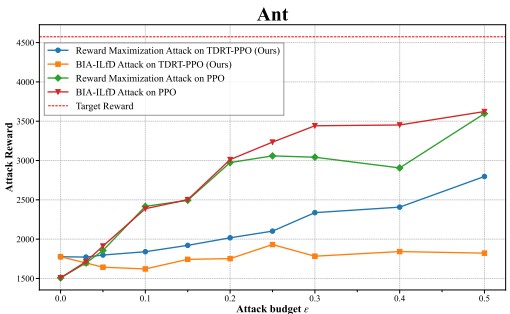 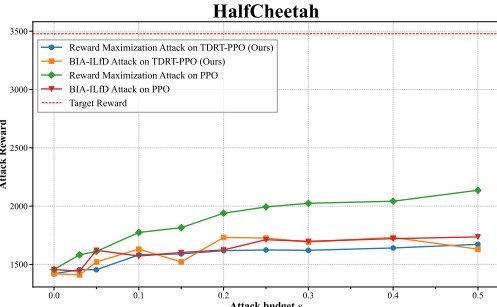

Figure 3: Evaluation on the Ant task.    Figure 4: Evaluation on the HalfCheetah task.

Figure 5: The horizontal axis represents the attack budget, which indicates the Adversary's intervention capability. The vertical axis shows the Attack Reward, which represents the reward obtained during the attack. Each value represents the average reward over 50 episodes.

regularization term, we employ either SGLD or convex relaxations of neural networks, similar to the approach used by (Zhang et al., 2020b). For convenience, we use $\hat{s} \in \mathcal{B}(s)$. However, since the defender does not have access to the $\mathcal{B}$ specified by the adversary, $\hat{s}$ is optimized to fall within a range specified by the defender.

# E    EFFICIENCY ANALYSIS

## E.1    EFFICIENCY OF BIA-ILFD/ILFO

We analyze the efficiency of BIA-ILfD/ILfO from two perspectives: training time of the adversarial policy and testing time. The training phase uses standard imitation learning without additional overhead, keeping learning costs low. Regarding demonstrations, the adversary's effort to prepare target policy demonstrations is more practical than designing new target reward functions. For testing efficiency, BIA only requires inference cost from the trained adversarial policy, giving it a significant advantage over methods like the targeted-PGD attack that requires optimization at each step. Additional experiments on attack performance are provided in Sections F.2 and F.3. Section F.2 investigates the relationship between demonstration quantity and attack performance, while Section F.3 analyzes attack efficiency across different attack budgets. These comprehensive experiments provide insights into both the data requirements and computational constraints of our proposed method.

## E.2    EFFICIENCY OF TDRT-PPO

TDRT-PPO adds regularization terms to standard RL algorithms. Following SA-PPO Zhang et al. (2020b), we applied common convex relaxation methods (Zhang et al., 2018; Wong & Kolter, 2018; Salman et al., 2019; Zhang et al., 2020a) to obtain KL term bounds. We implemented this using the auto_LiRPA toolkit(Xu et al., 2020), reducing regularization term computation costs and enabling efficient TDRT-PPO implementation. This regularization-based approach achieves shorter training times compared to adversarial training methods that require learning adversarial policies Liang et al. (2022).

# F    ADDITIONAL EXPERIMENTS

## F.1    PERFORMANCE EVALUATION IN MUJOCO

In this section, we conduct additional experiments on the Ant and HalfCheetah tasks from the OpenAI Gym Mujoco.

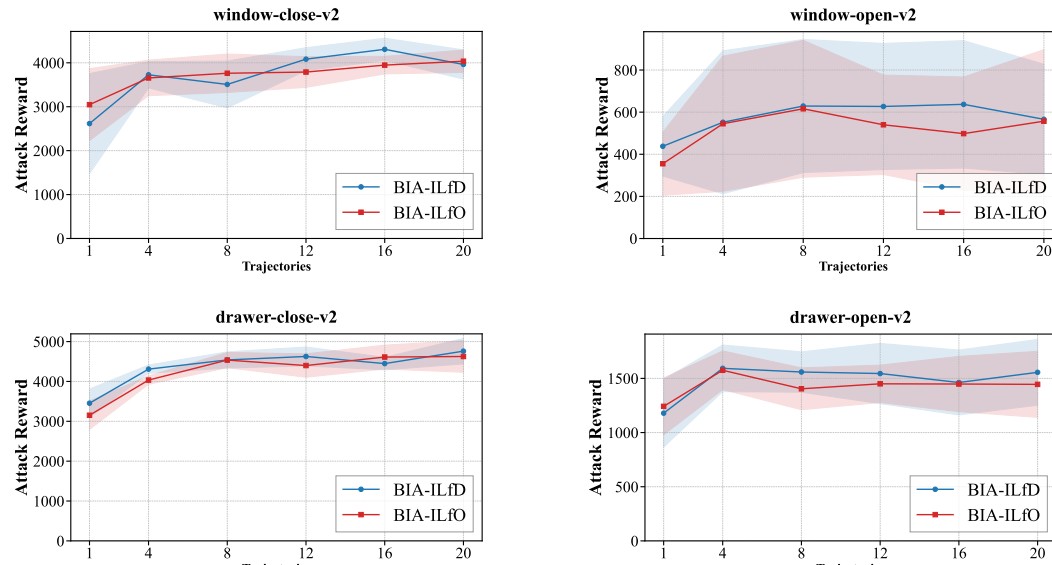

Figure 6: Attack performance of BIA-ILfD/ILfO with varying amounts of demonstrations. The x-axis shows the number of demonstration episodes, and the y-axis represents the Attack Reward. The attack budget $\epsilon$ is set to 0.3. Each environment name represents an adversarial task. The solid line and shaded area denote the mean and the standard deviation / 2 over 50 episodes.

**Setup** We set the adversary's goal as making a less-trained victim imitate the behavior of a well-trained target policy. In other words, the behavior-targeted attack is considered successful as the cumulative reward the victim obtains increases. We conduct experiments across several attack budgets $\epsilon$ to evaluate the performance of the attack.

**Results** The results are shown in Figures 5. A higher Attack Reward indicates a more successful attack. In the Ant task, we can see that the attacks against PPO are sufficiently successful. Additionally, we observe that TDRT-PPO has a lower Attack Reward than PPO. Thus, TDRT-PPO is more robust against behavior-targeted attacks. In the HalfCheetah task, we can confirm that the attacks are less effective than the Ant task. Furthermore, BIA fails to show effectiveness in the HalfCheetah task. This is likely due to the difficulty in imitating the target policy, which may have led to an eventual collapse in learning adversarial policies.

## F.2 ANALYSIS OF THE RELATIONSHIP BETWEEN DEMONSTRATIONS AND ATTACK PERFORMANCE OF BIA-ILFD/ILFO

In this section, we conduct additional experiments to investigate how the quantity and quality of target policy demonstrations affect the attack performance of BIA-ILfD/ILfO. We evaluate the performance of BIA-ILfD/ILfO using different amounts of demonstrations: 1, 4, 8, 12, 16, and 20 episodes. The victim is a vanilla PPO agent without any defense method. All experimental settings remain the same as in Section 7.2, with the attack budget $\epsilon$ set to 0.3.

Figure 6 shows the experimental results. Across all tasks, we observed no significant performance degradation when using up to 4 demonstration episodes. However, performance declined when only one demonstration episode was provided. This decline can be attributed to the variability in initial states, where the discriminator cannot effectively handle such diversity with extremely limited demonstrations.

We consider that the number of demonstrations required for successful behavior-targeted attacks depends on the environment's characteristics. In environments with deterministic state transitions and initial state distributions, where the target policy exhibits similar behavior across episodes, fewer demonstrations may suffice. Conversely, environments with more randomness in state transitions

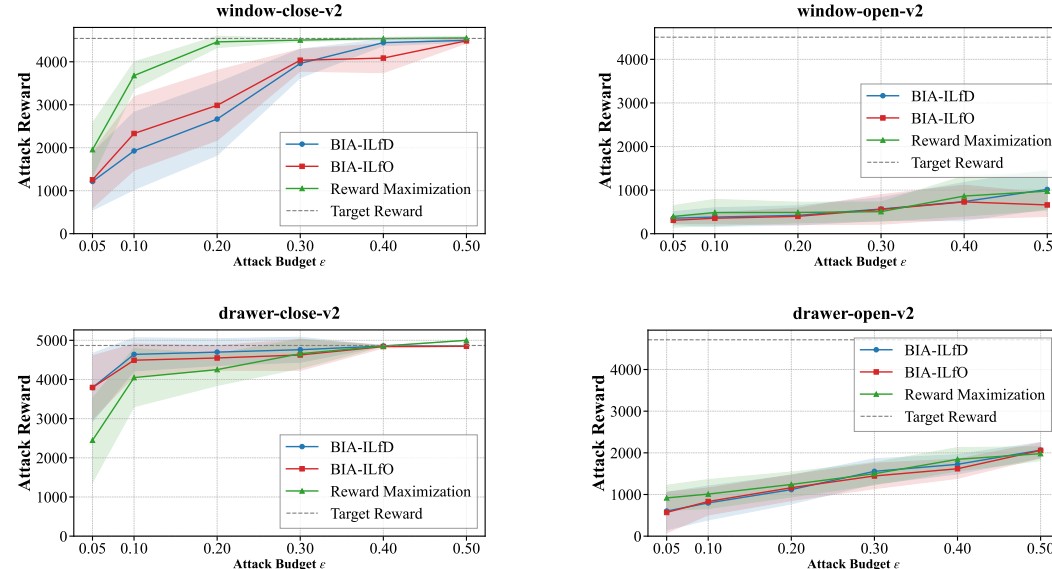

Figure 7: Attack performance of BIA-ILfD/ILfO with varying attack budget $\epsilon$. The x-axis shows the value of the attack budget, and the y-axis represents the Attack Reward. The Target Reward represents the cumulative reward obtained by the target policy and serves as the upper bound for the attack rewards of BIA-ILfD/ILfO. Each environment name represents an adversarial task. The solid line and shaded area denote the mean and the standard deviation / 2 over 50 episodes.

and initial state distributions require more demonstrations to ensure proper generalization of the discriminator.

### F.3 ANALYSIS OF THE RELATIONSHIP BETWEEN ATTACK BUDGET AND ATTACK PERFORMANCE OF BIA-ILFD/ILFO

In this section, we present a more comprehensive analysis of the attack efficiency of BIA-ILfD/ILfO. We conduct experiments across different attack budgets $\epsilon = 0.05, 0.1, 0.2, 0.3, 0.4, 0.5$ and evaluate the Attack Rewards. The victim is a vanilla PPO agent without any defense methods. All experimental parameters remain consistent with those in Section 7.2, and the results for $\epsilon = 0.3$ correspond to those presented in Table 1.

The experimental results are shown in Figure 7. Across all tasks, we observe that Attack Rewards increase proportionally as the attack budget increases. Notably, in the window-close and drawer-close tasks, the Attack Rewards nearly match the Target Rewards when given larger attack budgets, indicating that the attack successfully guides the victim to almost perfectly replicate the target behavior.

In the window-close and drawer-close tasks, where the attacks are particularly successful, we observed an interesting phenomenon. When the attack budget is small, there are high variances in Attack Rewards, but this variance decreases as the attack budget increases. The high variances indicate that among the 50 evaluation episodes, some attacks achieve perfect success while others completely fail. This finding suggests that the initial state significantly influences the attack success rate. We hypothesize that this occurs because certain initial states require the victim to perform actions that are rarely selected in their normal behavior, making the attack more challenging in these scenarios.

## G EXPERIMENTAL DETAILS

In this section, we provide the details of our experiments. We use a single NVIDIA Tesla V100 GPU and Intel Xeon E5-2698 v4 2.2GHz 20core CPU in all experiments.

---

**Algorithm 3** Optimization of adversarial states at each step in the targeted PGD attack

---

**Input:** Initial state $s$, target policy network $\pi_{\text{tgt}}$, victim policy network $\pi$, perturbation bound $\epsilon$, number of steps $T$

**Output:** Perturbed state $\hat{s}$

1: $\epsilon_{\text{step}} \leftarrow \epsilon/T$      ▷ Initialize perturbation step size
2: $s_{\min} \leftarrow s - \epsilon$, $s_{\max} \leftarrow s + \epsilon$      ▷ Set clipping bounds
3: $\hat{s} \leftarrow s + \text{Uniform}(-\epsilon_{\text{step}}, \epsilon_{\text{step}})$      ▷ Add initial random noise
4: $a_{\text{tgt}} \sim \pi_{\text{tgt}}(\hat{s}|\cdot)$      ▷ Compute target action
5: **for** $t = 1$ to $T$ **do**
6:      $a_{\text{current}} \sim \pi(\hat{s}|\cdot)$      ▷ Compute current victim's action
7:      $\mathcal{L} \leftarrow \|a_{\text{current}} - a_{\text{tgt}}\|_2^2$      ▷ Compute L2 loss
8:      $\hat{s} \leftarrow \hat{s} - \text{sign}(\nabla_{\hat{s}}\mathcal{L}) \cdot \epsilon_{\text{step}}$      ▷ Update state
9:      $\hat{s} \leftarrow \text{clip}(\hat{s}, s_{\min}, s_{\max})$      ▷ Clip state
10: **end for**
11: **return** $\hat{s}$

---

## G.1 ENVIRONMENTS

We conducted the experiments described in Section 7 using Meta-World(Yu et al., 2020), a benchmark that simulates robotic arm manipulation. All tasks in Meta-World share a common 39-dimensional continuous state space and a 4-dimensional continuous action space. In our experiments, we use four tasks: window-close, window-open, drawer-close, and drawer-open. The objective of each task is to move a specific object to a designated position.

**Reward Design** The reward design is specific to each task in MetaWorld. For example, the reward functions for window-close and window-open tasks are designed independently. Therefore, when attacking a victim trained on the window-close task to perform a window-open task, this differs from an untargeted attack since it does not simply minimize the victim's reward.

## G.2 THE DETAILS OF TARGETED PGD ATTACK

In this section, we provide a detailed explanation of targeted PGD attacks and present additional experiments. In Section G.2.1, we show the pseudo-code for targeted PGD attacks and provide specific details about their implementation. In Section G.2.2, we conduct experiments with various attack budgets and analyze the results to gain deeper insights into targeted PGD attacks.

### G.2.1 IMPLEMENTATION DETAILS OF THE TARGETED PGD ATTACK

We present the algorithm used to optimize the false state at each step for the targeted PGD attack in Algorithm 3. The attack aims to find optimal false states that minimize the difference between the victim's action and the target policy's action at each step. The algorithm performs $T$ iterations, during which it updates states using FGSM in each iteration. Specifically, it uses the L2 distance between current victim actions and target actions as the loss function and updates states by scaling in the direction of gradient signs by $\epsilon_{\text{step}}$. In all experiments, we set $T = 30$. We also implement random initialization for stable optimization.

### G.2.2 PERFORMANCE ANALYSIS OF TARGETED PGD ATTACKS UNDER DIFFERENT ATTACK BUDGETS

We evaluate the performance of targeted PGD attacks across various attack budgets $\epsilon$. We conduct attacks against victims trained with PPO without any defense method, using $\epsilon$ values of $[0.3, 0.5, 1.0, 3.0, 5.0, 10.0]$. The adversary's objectives are the same as in Section 7.

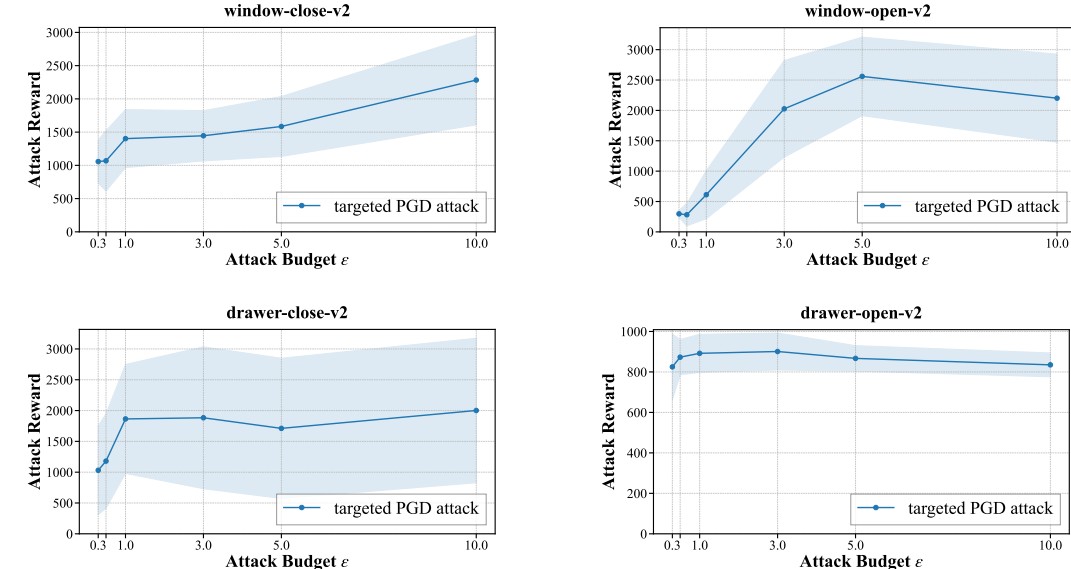

Figure 8: Performance of targeted PGD attacks under different attack budgets $\epsilon$. The x-axis represents the attack budget $\epsilon$, and the y-axis represents the Attack Reward. Each environment name represents an adversarial task. The solid line and shaded area denote the mean and the standard deviation / 2 over 50 episodes.

Figure 8 shows the experimental results. The environment names in the graphs represent adversarial tasks. For all tasks except drawer-open, we observed improved attack performance when the attack budget $\epsilon$ was set to very large values, such as 1.0 or higher. The window-open task showed particularly significant performance improvements. We believe this occurs because, with larger attack budgets, adversarial states that perfectly align the victim's actions with the target policy's actions become available at many steps, leading to overall behavioral alignment between the victim and target policy.

We attribute the failure of attacks, even at high attack budgets in the drawer-open task, to the victim never selecting actions that perfectly match the target policy, regardless of the state. In such environments, step-by-step optimization proves ineffective, suggesting that optimization across the entire episode would be more beneficial.

### G.3 FULL RESULTS IN DEFENSE PERFORMANCE EVALUATION

In our evaluation of defense methods in Section 7, we only provide the results of the best attack. In this section, we present the results of all attacks in 3. Targeted PGD is ineffective against robustly trained victims. We observed a trend where the Reward Maximization Attack tended to be slightly more effective than BIA when attacking highly robust victims. When the victim's robustness is high, it becomes difficult to make the victim behave like the target policy, which may cause the learning process in BIA to fail or collapse. On the other hand, in the Reward Maximization Attack, the reward serves as a good guidepost, allowing learning to proceed even when the victim's robustness is high.

### G.4 EFFECT OF REGULARIZATION ON PERFORMANCE

We investigated the impact of time-discounting regularization versus regular regularization on the Clean Rewards. Figure 9 shows the Clean Rewards when the regularization parameters are varied. TDRT-PPO (with time discounting) shows less performance degradation than SA-PPO (w/o time discounting). We can argue that this mitigation of performance degradation occurs because the upper bound in equation 13 becomes tighter due to time discounting.

Table 3: Comparison of Defense Methods. Each value is the average episode rewards ± standard deviation over 50 episodes. Clean Rewards are the rewards for the victim's tasks (no attack). Best Attack Rewards are the highest reward among the five types of adversarial attacks. The attack budget is set to $\epsilon = 0.3$.

| Adv Task | Methods | Clean Rewards (↑) | Attack Rewards (↓) | | | | | |
|---|---|---|---|---|---|---|---|---|
| | | | Random | Targeted PGD | Reward Maximization | BIA-ILfD (ours) | BIA-ILfO (ours) | Best Attack |
| window-close | PPO | 4508 ± 121 | 947 ± 529 | 1057 ± 652 | 4505 ± 65 | 3962 ± 666 | 4036 ± 510 | 4505 ± 65 |
| | ATLA-PPO (Zhang et al., 2021) | 4169 ± 467 | 1706 ± 1097 | 2028 ± 1387 | 4270 ± 188 | 2564 ± 787 | 3063 ± 1515 | 4270 ± 188 |
| | WocaR-PPO (Liang et al., 2022) | 2879 ± 1256 | 480 ± 5 | 480 ± 5 | 575 ± 135 | 381 ± 30 | 403 ± 29 | 575 ± 135 |
| | SA-PPO (Zhang et al., 2020b) | 4367 ± 107 | 478 ± 5 | 477 ± 5 | 485 ± 61 | 21 ± 12 | 21 ± 12 | 485 ± 61 |
| | TDRT-PPO (ours) | 4412 ± 55 | 422 ± 56 | 409 ± 44 | 482 ± 3 | 376 ± 43 | 377 ± 43 | **482 ± 3** |
| window-open | PPO | 4543 ± 39 | 322 ± 261 | 296 ± 117 | 506 ± 444 | 566 ± 523 | 557 ± 679 | 566 ± 523 |
| | ATLA-PPO (Zhang et al., 2021) | 4566 ± 80 | 354 ± 257 | 319 ± 250 | 547 ± 611 | 586 ± 649 | 532 ± 444 | 586 ± 649 |
| | WocaR-PPO (Liang et al., 2022) | 3645 ± 1575 | 287 ± 24 | 287 ± 24 | 295 ± 18 | 247 ± 64 | 253 ± 60 | 295 ± 18 |
| | SA-PPO (Zhang et al., 2020b) | 4092 ± 461 | 259 ± 46 | 258 ± 47 | 272 ± 37 | 200 ± 57 | 208 ± 60 | 272 ± 37 |
| | TDRT-PPO (ours) | 4383 ± 57 | 213 ± 54 | 213 ± 54 | 229 ± 54 | 254 ± 214 | 229 ± 137 | **254 ± 214** |
| drawer-close | PPO | 4714 ± 16 | 1069 ± 1585 | 1031 ± 1437 | 4658 ± 747 | 4760 ± 640 | 4626 ± 791 | 4760 ± 640 |
| | ATLA-PPO (Zhang et al., 2021) | 4543 ± 102 | 1004 ± 892 | 962 ± 1532 | 4858 ± 6 | 3919 ± 1808 | 3919 ± 1808 | 4858 ± 6 |
| | WocaR-PPO (Liang et al., 2022) | 4193 ± 304 | 562 ± 1335 | 834 ± 1635 | 3654 ± 1976 | 4867 ± 8 | 4838 ± 22 | 4867 ± 8 |
| | SA-PPO (Zhang et al., 2020b) | 2156 ± 453 | 3 ± 1 | 3 ± 1 | 3 ± 1 | 4 ± 2 | 4 ± 2 | 4 ± 2 |
| | TDRT-PPO (ours) | 4237 ± 93 | 1143 ± 1779 | 667 ± 1620 | 4770 ± 1 | 4860 ± 4 | 4860 ± 4 | 4860 ± 4 |
| drawer-open | PPO | 4868 ± 6 | 841 ± 357 | 825 ± 326 | 1499 ± 536 | 1556 ± 607 | 1445 ± 610 | 1556 ± 607 |
| | ATLA-PPO (Zhang et al., 2021) | 4863 ± 7 | 464 ± 270 | 421 ± 129 | 1158 ± 1026 | 831 ± 653 | 741 ± 561 | 1158 ± 1026 |
| | WocaR-PPO (Liang et al., 2022) | 4704 ± 654 | 410 ± 9 | 405 ± 8 | 579 ± 15 | 446 ± 28 | 442 ± 22 | 579 ± 15 |
| | SA-PPO (Zhang et al., 2020b) | 4161 ± 1537 | 368 ± 9 | 368 ± 9 | 368 ± 9 | 403 ± 49 | 403 ± 49 | 403 ± 49 |
| | TDRT-PPO (ours) | 4802 ± 27 | 378 ± 10 | 378 ± 10 | 378 ± 10 | 357 ± 4 | 357 ± 4 | **378 ± 10** |

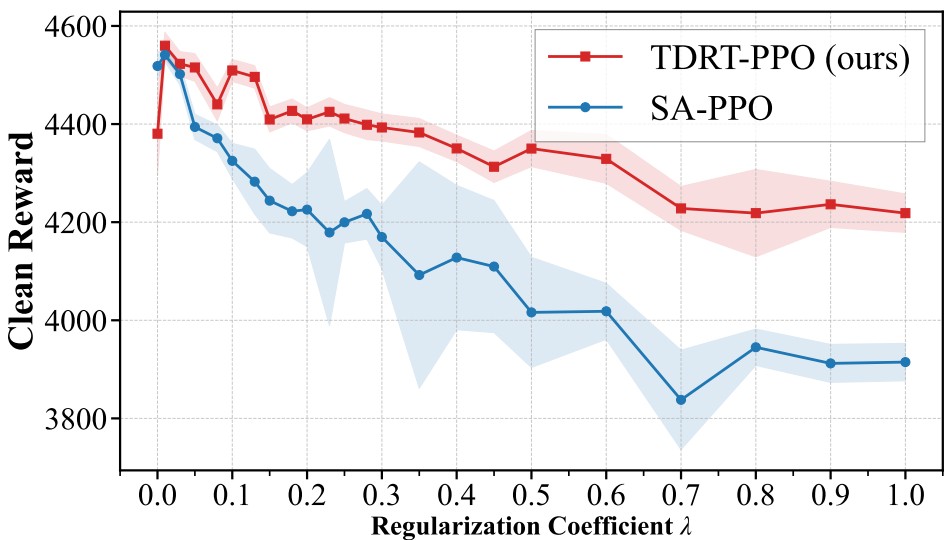

Figure 9: Regularization effect on performance

