# OpenReview forum: "Robust Deep Reinforcement Learning against ADVERSARIAL BEHAVIOR MANIPULATION"
_ICLR.cc/2025/Conference — Submitted to ICLR 2025_

### Official Review · Reviewer_kKvm · 2024-10-23

**Soundness:** 3
**Presentation:** 3
**Contribution:** 3
**Rating:** 6
**Confidence:** 3

**Summary:**

This paper first proposes a method to mount a targeted black-box attack to adversarially manipulate a victim RL policy towards a target policy. Unlike previous works, the proposed method, leverages imitation learning and is extendable to the no-box setting and does not require environment heuristics or the creation of a target policy, but only the trajectories of the target policy is required. Additionally, the authors also developed a defense framework that is specific towards behavioral manipulation by introducing a time-discounted regularization to induce robustness of the victim policy towards adversarial manipulation while maintaining nominal performance. Experiments were shown on MuJoCo and MetaWorld environments.

**Strengths:**

1. Overall, the paper is relatively well written, clear and easy to follow.

2. The problem of adversarial manipulation towards a targeted behavior and the corresponding need to defend against such attacks are not necessarily new, but I believe the way the authors formulate the adversarial problem and derive corresponding algorithm and the formulation of the time-discounted regularization is original, to the best of my knowledge.

3. In terms of significance, I believe that this area of research (adversarial manipulation and defense for RL) is a niche (more on that below) but important area of study. More importantly, I believe any advancement in this area could also lead to advancement in other areas of RL robustness.

**Weaknesses:**

1. The main motivation for behavior manipulation and claim that it is different from reward minimization from adversarial point of view is a little challenging to imagine in the real world.

2. Similarly, I'm not convinced that the idea of time-discounted regularization is a generalizable method of defense.

3. The results shown in the main paper are convincing but somewhat limited and the results in MuJoCo shown in the Appendix are somewhat mixed, which further weakens the paper's claims.

**Questions:**

1. One of the main differentiation of this paper from existing works that the authors claimed are that behavioral manipulation is different from naive adversarial attacks that seeks to minimize rewards. While I understand this conceptually, I find it hard to imagine a situation in a real world where adversarial behavior manipulation does not somehow lead to the same scenario as reward minimization. Could the authors provide some concrete examples?

2. Related to the question above, in the contributions, the authors wrote: "TDRT is robust against attacks that do not significantly reduce rewards but greatly alter behavior". Does this assume that there are environments where rewards are not strongly correlated with sub-optimal behavior? In other words, is there an assumption that the reward function still returns a relatively high reward when the policy is suboptimal?

3. On the topic of time-discounted regularization, could the authors comment on what happens in scenarios where the RL is non-episodic or is facing an infinite horizon problem? I can understand if the authors claim that this method works for situations where the initial time steps are critical, but there are also many problems where the actions and state are equally critical in the later stages of the episode.

4. For the attack algorithm, the authors mentioned that a target policy is not needed, only access to the trajectory of the target policy is needed. In such cases when target policy is not available, is there a specific amount of trajectory that is needed for the attack to be mounted successfully?

5. In Equation 12. the formulation suggest that the loss function minimizes the victim's behavior even under the worse-case adversarial policy. How is the worse-case adversarial policy defined here?  Isn't the premise of the attack a policy that doesn't necessarily minimizes the reward but rather pushes the victim towards a target behavior?

6. In the results section for Defense, could the authors explain why the clean rewards represent rewards when the victim is trained to achieve the adversarial task without attacks? Shouldn't the clean rewards represent the victim's performance for the nominal task without any attack and wouldn't a comparison showing how the naive and adversarially trained victim policy performs in the a normal and attacks setting makes more sense? Please let me know if I misunderstand this section.

7. As mentioned in the section above, additional results on more task in the MetaWorld / MuJoCo showing the effectiveness of the proposed attack and defense method would definitely make the results more convincing.

---

> ### Author Response · Authors · 2024-11-18
> **Response to Reviewer kKvm (Part 1)**
>
> We thank the reviewer for their valuable feedback. Below, we provide clarifications:
>
> > W1. The main motivation for behavior manipulation and claim that it is different from reward minimization from adversarial point of view is a little challenging to imagine in the real world.
>
> > Q1. One of the main differentiation of this paper from existing works that the authors claimed are that behavioral manipulation is different from naive adversarial attacks that seeks to minimize rewards. While I understand this conceptually, I find it hard to imagine a situation in a real world where adversarial behavior manipulation does not somehow lead to the same scenario as reward minimization. Could the authors provide some concrete examples?
>
> Thank you for this important question about real-world implications. We would like to provide a concrete example to illustrate the difference between behavior manipulation and reward minimization.
> Consider a real-world example involving autonomous vehicles or mobile robots reaching specific destinations. An adversary can force the victim to take a particular route. In this case, since the victim still reaches its final destination, there is no significant reduction in the accumulated reward.
> However, by forcing the victim to take a particular route, the adversary can carry out various attacks, such as launching additional attacks at a location prepared in advance or causing traffic congestion. These scenarios are a serious threat but fundamentally different from reward minimization attacks. For further motivating examples, please refer to response 1 to Reviewer M1ng.
>
> > Q2. Related to the question above, in the contributions, the authors wrote: "TDRT is robust against attacks that do not significantly reduce rewards but greatly alter behavior". Does this assume that there are environments where rewards are not strongly correlated with sub-optimal behavior? In other words, is there an assumption that the reward function still returns a relatively high reward when the policy is suboptimal?
>
> Yes, there are environments where significant changes in behavior do not necessarily lead to significant reductions in reward, as in our above example of forcing specific route navigation.
> Furthermore, the effectiveness of defenses against behavior-targeted attacks is not limited to such environments. In tasks where significant behavioral changes lead to substantial reward reduction, defenses against behavior-targeted attacks will also be effective against reward minimization attacks.
>
> > W2. Similarly, I'm not convinced that the idea of time-discounted regularization is a generalizable method of defense.
>
> > Q3. On the topic of time-discounted regularization, could the authors comment on what happens in scenarios where the RL is non-episodic or is facing an infinite horizon problem? I can understand if the authors claim that this method works for situations where the initial time steps are critical, but there are also many problems where the actions and state are equally critical in the later stages of the episode.
>
> Thank you for this important question about the generalizability of time-discounted regularization.
> The defense method against behavior-targeted attacks aims to minimize changes in the victim's behavior, not the victim's cumulative reward. Therefore, we believe that this method remains effective in non-episodic and infinite horizon problems.
> As you correctly point out, there are many environments where the later stages of episodes are critical from the perspective of the victim's cumulative reward. However, as shown in Theorem 1, we claim that the early stages of episodes are particularly important from the perspective of overall behavioral change in general, regardless of environment.

---

> ### Author Response · Authors · 2024-11-18
> **Response to Reviewer kKvm (Part 2)**
>
> > Q4. For the attack algorithm, the authors mentioned that a target policy is not needed, only access to the trajectory of the target policy is needed. In such cases when target policy is not available, is there a specific amount of trajectory that is needed for the attack to be mounted successfully?
>
> Thank you for this important question about the trajectory requirements of our attack method.
> We consider that the number of demonstrations required for successful behavior-targeted attacks depends on the environment's characteristics. In environments with deterministic state transitions and initial state distributions, where the target policy exhibits similar behavior across episodes, fewer demonstrations may suffice. Conversely, environments with more randomness in state transitions and initial state distributions require more demonstrations to ensure proper generalization of the discriminator.
>
> To further analyze the trajectory requirements, we conducted additional experiments to examine the relationship between the amount of trajectory data and attack performance, which we have included in the Appendix. While we used 20 episode trajectories in our experiments in Section 7, our additional experiments show that comparable attack performance can be achieved with as few as 4 episode trajectories. Regarding experimental evaluation of the number of expert trajectories and attack efficiency, please refer to response 2 of Reviewer JMYu, too.
>
> > Q5. In Equation 12. the formulation suggest that the loss function minimizes the victim's behavior even under the worse-case adversarial policy. How is the worse-case adversarial policy defined here? Isn't the premise of the attack a policy that doesn't necessarily minimizes the reward but rather pushes the victim towards a target behavior?
>
> In Equation 12, the worst-case adversarial policy is defined as an adversary that maximizes the change in the victim's behavior, specifically maximizing the difference in state-action distribution.
> As you correctly point out, considering an adversary that forces specific behaviors on the victim would be most appropriate. However, since the defender cannot access the target policy specified by the adversary during the victim policy learning process, we have adopted this definition of the worst-case adversary.
>
> > Q6. In the results section for Defense, could the authors explain why the clean rewards represent rewards when the victim is trained to achieve the adversarial task without attacks? Shouldn't the clean rewards represent the victim's performance for the nominal task without any attack and wouldn't a comparison showing how the naive and adversarially trained victim policy performs in the a normal and attacks setting makes more sense? Please let me know if I misunderstand this section.
>
> We apologize for any confusion our explanation may have caused, and we would like to provide a clearer definition of clean rewards.
> The clean rewards represent the victim's performance for the nominal task without any attack, not the reward in the adversarial task. To illustrate with a concrete example, when an adversary aims to make the victim perform the window-open task, the clean reward represents the reward in the window-close task that the victim was originally trained for. We have updated the explanation of the clean reward.
>
> > W3. The results shown in the main paper are convincing but somewhat limited and the results in MuJoCo shown in the Appendix are somewhat mixed, which further weakens the paper's claims.
>
> > Q7. As mentioned in the section above, additional results on more task in the MetaWorld / MuJoCo showing the effectiveness of the proposed attack and defense method would definitely make the results more convincing.
>
> Thank you for suggesting the need for additional experimental validation. We appreciate this constructive feedback. We are currently conducting experiments on additional environments.

---

> > ### Comment · Reviewer_kKvm · 2024-11-23
> > **Response to rebuttal**
> >
> > >Thank you for this important question about real-world implications. We would like to provide a concrete example to illustrate the difference between behavior manipulation and reward minimization. Consider a real-world example involving autonomous vehicles or mobile robots reaching specific destinations. An adversary can force the victim to take a particular route. In this case, since the victim still reaches its final destination, there is no significant reduction in the accumulated reward. However, by forcing the victim to take a particular route, the adversary can carry out various attacks, such as launching additional attacks at a location prepared in advance or causing traffic congestion. These scenarios are a serious threat but fundamentally different from reward minimization attacks. For further motivating examples, please refer to response 1 to Reviewer M1ng.
> >
> > These are definitely valid scenarios and I thank the authors for clarifying this point. However, typically even in routing scenarios, the reward isn't necessarily dependent on just the final destination, but also typically takes into account the cost of the route. Perhaps it is more important to clarify the assumption that the method is specifically for situations where a change in behavior does not correlate to the reward.
> >
> > I have no other comments towards the other responses. Please do update the draft to incorporate some of the responses in order to clarify the paper.
> >
> > I will currently maintain my score (pending additional experiments)

---

> ### Author Response · Authors · 2024-11-25
> **Response to Review Comments - Additional Experimental Results (Part 1)**
>
> Dear Reviewer kKvm,
>
> Thank you for your thoughtful response. We agree that our defense is particularly effective in environments where there is no correlation between behavior and rewards. We have incorporated this point, along with our other responses to the reviews, into the paper.
>
> To demonstrate the effectiveness of our method more comprehensively, we have conducted additional experiments in new environments. In MetaWorld, we have performed experiments on {faucet-close, faucet-open}, {handle-press-side, handle-pull-side}, and {door-lock, door-unlock}. We have also conducted experiments in the MuJoCo Hopper environment.
>
> ## Attack Results
>
> | Adv Task | Target Reward | Random | Targeted PGD (Upper bound  | Reward Maximization | BIA-ILfD(Ours) | BIA-ILfO(Ours) |
> | --- | --- | --- | --- | --- | --- | --- |
> | faucet-close | *4754 $\pm$ 15* | 897 $\pm$ 171 | 1092 $\pm$ 192 | **3409 $\pm$ 652** | 3316 $\pm$ 648 | 3041 $\pm$ 502 |
> | faucet-open | *4544 $\pm$ 800* | 1372 $\pm$ 81 | 1414 $\pm$ 86 | 1448 $\pm$ 64 | **3031 $\pm$ 1493** | 2718 $\pm$ 1293 |
> | handle-press-side | *4546 $\pm$ 721* | 1865 $\pm$ 1340 | 1994 $\pm$ 1225 | 4625 $\pm$ 175 | **4631 $\pm$ 408** | 4627 $\pm$ 586 |
> | handle-pull-side | *4442 $\pm$ 732* | 1426 $\pm$ 1617 | 1398 $\pm$ 1524 | 3617 $\pm$ 1363 | **4268 $\pm$ 740** | 4193 $\pm$ 517 |
> | door-lock | *3845 $\pm$ 79* | 589 $\pm$ 494 | 640 $\pm$ 664 | 1937 $\pm$ 1186 | **2043 $\pm$ 1229** | 1906 $\pm$ 1045 |
> | door-unlock | *4690 $\pm$ 33* | 391 $\pm$ 59 | 431 $\pm$ 61 | **703 $\pm$ 220** | 563 $\pm$ 220 | 590 $\pm$ 228 |
>
> We show the attack results on vanilla PPO, which does not use any defense methods. Our proposed method proves effective in all tasks except door-unlock. Particularly high attack rewards are observed in the faucet-close/open and handle-press/pull-side tasks. However, the attacks are less successful in the door-unlock tasks.
>
> We believe the variation in attack performance across environments stems from a fundamental limitation: the adversary cannot force the victim to take actions that it rarely selects. This tendency can be observed in the attack rewards from Random Attacks. When Random Attack rewards are high, it indicates that the two opposite task pairs exhibit similar behaviors, making it easier for the victim to adopt the adversary's desired behavior. Conversely, when Random Attack rewards are low, it suggests that the opposite task pairs have entirely different behaviors, making it difficult to find false states that could induce the adversary's desired behavior.
>
> ## Defense Results
>
> | Adv Task | Victim Method | Clean Reward ($\uparrow$) | Best Attack Reward ($\downarrow$)|
> | --- | --- | --- | --- |
> | faucet-close | Vanilla PPO | 4544 $\pm$ 800 | 3409 $\pm$ 652 |
> |  | SA-PPO | 4304 $\pm$ 42 | 1558 $\pm$ 406 |
> |  | **TDRT-PPO (Ours)** | 4740 $\pm$ 17 | 1580 $\pm$ 234 |
> | faucet-open | Vanilla PPO | 4754 $\pm$ 15 | 3031 $\pm$ 1493 |
> |  | SA-PPO | 4380 $\pm$ 43 | 1763 $\pm$ 255 |
> |  | **TDRT-PPO (Ours)** | 4630 $\pm$ 11 | 1942 $\pm$ 261 |
> | handle-press-side | Vanilla PPO | 4442 $\pm$ 732 | 4631 $\pm$ 408 |
> |  | SA-PPO | 3226 $\pm$ 806 | 2193 $\pm$ 401 |
> |  | **TDRT-PPO (Ours)** | 4321 $\pm$ 215 | 2276 $\pm$ 124 |
> | handle-pull-side | Vanilla PPO | 4546 $\pm$ 721 | 4268 $\pm$ 740 |
> |  | SA-PPO | 4094 $\pm$ 350 | 1532 $\pm$ 167 |
> |  | **TDRT-PPO (Ours)** | 4468 $\pm$ 126 | 1423 $\pm$ 84 |
> | door-lock | Vanilla PPO | 4690 $\pm$ 33 | 2043 $\pm$ 1229 |
> |  | SA-PPO | 2299 $\pm$ 1491 | 478 $\pm$ 7 |
> |  | **TDRT-PPO (Ours)** | 2769 $\pm$ 1411 | 487 $\pm$ 11 |
> | door-unlock | Vanilla PPO | 3845 $\pm$ 79 | 703 $\pm$ 220 |
> |  | SA-PPO | 2017 $\pm$ 497 | 701 $\pm$ 403 |
> |  | **TDRT-PPO (Ours)** | 3680 $\pm$ 290 | 691 $\pm$ 356 |
>
> We also provide the defense results on SA-PPO and TDRT-PPO. Regarding defense performance, TDRT-PPO demonstrates effectiveness in most tasks, similar to the main results. In all tasks, TDRT-PPO achieves higher Clean Rewards than SA-PPO while maintaining equivalent robustness. In the door-lock task, both SA-PPO and TDRT-PPO show decreased Clean Rewards, suggesting that regularization might not be effective in this environment.

---

> > ### Author Response · Authors · 2024-11-25
> > **Response to Review Comments - Additional Experimental Results (Part 2)**
> >
> > Next, we present the results from the MuJoCo environment. The adversary's objective in this setting is to make the less trained victim perform tasks effectively by using a well-trained policy as the target policy.
> >
> > ## Vanilla PPO Results
> >
> > Victim: Vanilla PPO (less-trained), Original Reward: 1405 $\pm$ 54, Target Reward: 3513 $\pm$ 182
> >
> > | Method | $\epsilon=$0.1 | $\epsilon=$0.2 | $\epsilon=$0.3 | $\epsilon=$0.4 | $\epsilon=$0.5 |
> > | --- | --- | --- | --- | --- | --- |
> > | Reward Maximization | 2024 $\pm$ 481 | 2398 $\pm$ 7 | 3056 $\pm$ 42 | 3153 $\pm$ 8 | 3357 $\pm$ 516 |
> > | BIA-ILfD | 1504 $\pm$ 0 | 1918 $\pm$ 7 | 3198 $\pm$ 9 | 3420 $\pm$ 4 | 3414 $\pm$ 6 |
> >
> > ## TDRT-PPO Results
> >
> > Victim: TDRT-PPO (less-trained), Original Reward: 1310 $\pm$ 191, Target Reward: 3513 $\pm$ 182
> >
> > | Method | $\epsilon=$0.1 | $\epsilon=$0.2 | $\epsilon=$0.3 | $\epsilon=$0.4 | $\epsilon=$0.5 |
> > | --- | --- | --- | --- | --- | --- |
> > | Reward Maximization | 1085 $\pm$ 543 | 1709 $\pm$ 337 | 2190 $\pm$ 234 | 2225 $\pm$ 211 | 2433 $\pm$ 365 |
> > | BIA-ILfD | 1210 $\pm$ 300 | 1061 $\pm$ 380 | 1481 $\pm$ 2 | 2011 $\pm$ 8 | 1944 $\pm$ 284 |
> >
> > In the Hopper environment, while we have confirmed sufficient attack effectiveness, the defense provided by TDRT-PPO, while effective, shows less complete effectiveness compared to the MetaWorld environment. We attribute this to the similarity between the target policy and the victim's policy behaviors in this attack setting. Whereas in MetaWorld, the goal is to make the victim perform opposite tasks, in MuJoCo, the goal is to make an undertrained victim imitate a well-trained target policy. Since both the victim and target policy are trained on the same task, some attack success is expected, even with regularization.

---

> > > ### Comment · Reviewer_kKvm · 2024-11-26
> > > **Response to additional results**
> > >
> > > I would like to thank the authors for their effort in conducting additional experiments and clarifying the results.
> > >
> > > While I would be willing to raise the score to a 7 if the option was available, unfortunately I don't believe the existing results qualify for a score for of 8, due to the fact that there's still too many "if's and else's" that the methods needs to be conditioned on to be effective.
> > >
> > > As such, I am maintaining my score as a 6. All the best and I hope the authors could include the discussions above into their paper.

---

> ### Author Response · Authors · 2024-11-28
> **Thank you for your valuable comments**
>
> ### We sincerely thank you for the time and effort you have dedicated to reviewing our work.
>
> We are truly grateful for your excellent feedback and positive evaluation of our research. We will carefully compile our experimental results and include these discussions in the camera-ready version. With this appreciation in mind, we humbly believe that **the variability in effectiveness stems from the inherent nature of behavior-targeted attacks rather than limitations in our proposed method.**
>
> The most significant constraint of behavior-targeted attacks lies in the fundamental difficulty of inducing behaviors that victims rarely exhibit. As a result, the difficulty of attacks varies greatly across environments, and environments where our method appears less effective are precisely those where attacks are inherently very challenging. For example, in the door-unlock environment, while our attacks showed limited success, the reward in Random Attack was also low, indicating a substantial gap between the victim's behavior and the target policy's behavior.
>
> In environments where our proposed method failed to succeed, all other baseline methods also failed. This suggests that these specific environments present fundamental challenges to such attacks. Therefore, the variability in results reflects the nature of the problem space rather than the limitations in our approach.
>
> Once again, we deeply appreciate your constructive feedback over multiple rounds and the time and effort you have invested in our work.

---

### Official Review · Reviewer_uL7G · 2024-11-01

**Soundness:** 2
**Presentation:** 3
**Contribution:** 4
**Rating:** 5
**Confidence:** 4

**Summary:**

The paper provides formalizations for a previously uncovered adversarial RL scenario in which an adversary is directing the victim toward some specific target behavior. Under this formulation, the paper proposes a robust policy regularizer based on the F-divergence between the policy and the composite policy+adversary. Empirical results demonstrate the effectiveness of the attacks and defenses provided.

**Strengths:**

The attention to behavior-manipulating attacks is very novel. Although the concept has been mentioned before, this work is the first to provide formalizations and a solution.

The problem setup and theoretical contributions are clear.

The attention to both adversarial and robust angles of the problem makes a thorough contribution.

**Weaknesses:**

A few baselines are missing which should be compared to. For defenses, there are methods [1] and [2]  and [3] that should be considered versus TDRT. For attacks, the PA-AD component of [3] should be compared to BIA-ILfD/O.

The empirical setup is unclear, as is the presentation of the results in Table 1. The information in Table 1 is confusing, for instance, the attack rewards appear to be higher-is-better (assuming boldface = best), but the boldface scores are lower than the target reward. This is more confusing when compared to Table 2, which is clearly lower-is-better for the attacks.

The description of experimental tasks is lacking--from section 7.1, it is unclear how rewards are distributed for each goal, and how they correlate to agent performance.

*[1] Xiangyu Liu, Chenghao Deng, Yanchao Sun, Yongyuan Liang, Furong Huang: Beyond Worst-case Attacks: Robust RL with Adaptive Defense via Non-dominated Policies. ICLR 2024* \
*[2] Roman Belaire, Pradeep Varakantham, Thanh Hong Nguyen, David Lo: Regret-based Defense in Adversarial Reinforcement Learning. AAMAS 2024: 2633-2640*\
*[3] Yanchao Sun, Ruijie Zheng, Yongyuan Liang, Furong Huang: Who Is the Strongest Enemy? Towards Optimal and Efficient Evasion Attacks in Deep RL. ICLR 2022*

**Questions:**

1) How do the opposing goals in the tasks distribute rewards? Is one of the states (i.e. window-open) a positive reward, and the opposite negative? If that is the case, this seems identical to classic reward-min attacks, unless a formalization can be provided to differentiate the two scenarios.

2) Related to question 1, is there a way to formalize the selection process for $\pi_{tgt}$? I understand the novelty of the attack type in isolation, however, the only well-defined $\pi_{tgt}$ would be min-reward, which is the existing attack paradigm.

3) In equations 8 and 11, my understanding is that the optimization only depends on $\nu$ unless D interacts with $\pi$ in some way. Since $\pi$ is a fixed policy, why can't equation 8 be solved directly? The problem seems easier than if $\pi$ was dynamic.

4) The authors mention that there are no defense methods currently for behavior-targeted attacks, so untargeted defense methods are used. What considerations are made to adapt these methods to this setting? For instance, is the target reward used as the learning objective for PA-ATLA, or a different notion of reward?

Minor comments:\
-$\hat{p}$ is defined in eq 10 but unused in eq 9.\
-Some misspellings throughout, like the "Notaion" header in Section 3.

---

> ### Author Response · Authors · 2024-11-18
> **Response to Reviewer uL7G (Part 1)**
>
> We thank the reviewer for their valuable feedback. Below, we provide clarifications:
>
> > W1. A few baselines are missing which should be compared to. For defenses, there are methods [1] and [2] and [3] that should be considered versus TDRT. For attacks, the PA-AD component of [3] should be compared to BIA-ILfD/O.
>
> We appreciate the references given and agree that they are relevant to our work. We are currently conducting experiments with these baselines. However, please note that these approaches tackle a different problem than the one we consider, and the PA-AD component of [3] is not applicable to our setting. In what follows, we are going to explain how [1,2,3] are different from our work.
>
> The key difference between methods [1,2,3] and our proposed defense lies in their objectives. While [1,2,3] aims to minimize the decrease in the victim's cumulative reward, our approach focuses on minimizing changes in the victim's behavior. In our experiments, we used ATLA-PPO and WocaR-PPO as baselines that aim to minimize cumulative reward decrease.
>
> There are differences between PA-AD and our proposed attack method in terms of access to the victim's policy. PA-AD requires white-box access to the victim's policy and a reward function for learning the target policy. In contrast, BIA-ILfD/O can be applied with black-box/no-box access to the victim's policy and only requires trajectories of the target policy rather than a reward function.
>
> > W2. The empirical setup is unclear, as is the presentation of the results in Table 1. The information in Table 1 is confusing, for instance, the attack rewards appear to be higher-is-better (assuming boldface = best), but the boldface scores are lower than the target reward. This is more confusing when compared to Table 2, which is clearly lower-is-better for the attacks.
>
> Thank you for pointing out the confusion in our presentation of results. We have improved the clarity of our explanations about Table 1.
> The target reward represents the cumulative reward of the expert in the adversarial task and serves as an upper bound for the attack reward. In Table 1, when comparing attack performance, we bold the highest attack reward among the attack methods.
> However, we recognize that this presentation might cause confusion when compared with the defense performance results in Table 2. To address this, we have updated the table caption to include a clear explanation of the target reward and revised the overall table design to improve comprehension.
>
> > W3. The description of experimental tasks is lacking--from section 7.1, it is unclear how rewards are distributed for each goal, and how they correlate to agent performance.
>
> > Q1. How do the opposing goals in the tasks distribute rewards? Is one of the states (i.e. window-open) a positive reward, and the opposite negative? If that is the case, this seems identical to classic reward-min attacks, unless a formalization can be provided to differentiate the two scenarios.
>
> The reward design is specified for each task in Metaworld. All tasks, including window-open and window-close tasks, have their own distinct reward functions defined within Metaworld. For example, the reward functions for “window-open” and “window-close” are separately defined for their goals and not defined by not defined by the inversion of the sign.
> Therefore, maximizing the reward in the window-close task is not equivalent to minimizing the reward in the window-open task, making it fundamentally different from conventional reward-min attacks. We have added detailed explanations of the reward design for these tasks to Section G.1.
>
> > Q2. Related to question 1, is there a way to formalize the selection process for πtgt? I understand the novelty of the attack type in isolation, however, the only well-defined πtgt would be min-reward, which is the existing attack paradigm.
>
> For reasons explained above, reward design in Metaworld is task-dependent, $\pi_{\text{tgt}}$ does not correspond to a reward-minimizing policy.
>
> [1] Xiangyu Liu, Chenghao Deng, Yanchao Sun, Yongyuan Liang, Furong Huang: Beyond Worst-case Attacks: Robust RL with Adaptive Defense via Non-dominated Policies. ICLR 2024
>
> [2] Roman Belaire, Pradeep Varakantham, Thanh Hong Nguyen, David Lo: Regret-based Defense in Adversarial Reinforcement Learning. AAMAS 2024: 2633-2640
>
> [3] Yanchao Sun, Ruijie Zheng, Yongyuan Liang, Furong Huang: Who Is the Strongest Enemy? Towards Optimal and Efficient Evasion Attacks in Deep RL. ICLR 2022

---

> ### Author Response · Authors · 2024-11-18
> **Response to Reviewer uL7G (Part 2)**
>
> > Q3. In equations 8 and 11, my understanding is that the optimization only depends on $\nu$ unless D interacts with $\pi$ in some way. Since $\pi$ is a fixed policy, why can't equation 8 be solved directly? The problem seems easier than if $\pi$ was dynamic.
>
> Thank you for this insightful question about the optimization in equations 8 and 11. We would like to clarify the optimizations.
> The reward functions in equations 8 and 11 are defined for $\pi\circ\nu$, which makes it impossible to optimize directly for $\nu$ alone. If the reward function were defined directly for the adversarial policy (i.e., determined by states, falsified states, and next states $R(s, \hat{s}, s’)$), it could be optimized as a reinforcement learning problem for the adversarial policy.
> Lemma 1 shows that from the reward functions for $\pi\circ\nu$ given in equations 8 and 11, we can design a reward function for $\nu$ in a newly defined MDP. Therefore, our proposed method utilizes Lemma 1 to enable the learning of the adversarial policy.
>
> > Q4. The authors mention that there are no defense methods currently for behavior-targeted attacks, so untargeted defense methods are used. What considerations are made to adapt these methods to this setting? For instance, is the target reward used as the learning objective for PA-ATLA, or a different notion of reward?
>
> Thank you for this important question about adapting untargeted defense methods.
> We apply existing defense methods against untargeted attacks directly.
> As you correctly point out, using the target reward in adversarial training for methods like ATLA-PPO and PA-ATLA-PPO might improve robustness against behavior-targeted attacks. However, this approach would not be realistic because defenders cannot access the target reward, which the adversary uses during the victim policy learning. Therefore, we do not use the target reward in our adversarial training process.
>
> > -$\hat{p}$ is defined in eq 10 but unused in eq 9.
> -Some misspellings throughout, like the "Notaion" header in Section 3.
>
> Thank you for catching these errors. We appreciate your careful review of our manuscript. We have corrected these misspellings and updated the paper accordingly. It is intentional that $\hat{p}$ is defined in eq 10. is not used in eq 9. $\hat{p}$ serves as the transition function in the MDP $\hat{M}$.

---

> > ### Comment · Reviewer_uL7G · 2024-11-26
> >
> > Thanks for your thorough response to my concerns. The major concern I had w.r.t. novelty is resolved, though I do have a few others regarding baselines:
> >
> > > While [1,2,3] aims to minimize the decrease in the victim's cumulative reward, our approach focuses on minimizing changes in the victim's behavior. In our experiments, we used ATLA-PPO and WocaR-PPO as baselines that aim to minimize cumulative reward decrease.
> >
> > To clarify, are the authors stating that [1,2,3] and ATLA-PPO/WocaR-PPO belong to the same group of methodologies?
> > Additionally, a minor note on pedantics: in the paper on lines 458-459, WocaR-PPO is stated to be an adversarial retraining method. To my understanding, the method maximizes a computed lower bound on reward, rather than interacting with an adversary at training time.
> >
> > > PA-AD requires white-box access to the victim's policy and a reward function for learning the target policy.
> >
> > > [ATLA-PPO, PA-ATLA-PPO] would not be realistic because defenders cannot access the target reward, which the adversary uses during the victim policy learning. Therefore, we do not use the target reward in our adversarial training process.
> >
> > Thank you for the clarification. I understand the novelty of the work makes it difficult or impossible to find 1:1 baselines to compare to. However, given that the PGD attacks and Target Reward-Max attacks used in Table 1 each utilize a property in the constraints quoted above, a comparison to white-box PA-AD would make the experiments more thorough, as it is a SOTA method related directly to ATLA-PPO.
> >
> > >  The reward design is specified for each task in Metaworld.
> >
> > Thank you for specifying this. Given that there are several tasks per environment in Metaworld, would a multi-objective agent be something to consider as a robust alternative? I am less familiar with the specifics of multi-objective RL, but it would be an interesting comparison, something to the effect of: $R_{MO}(\cdot) = R(\cdot) - R_{tgt}(\cdot)$.
> >
> > > > Q2. Related to question 1, is there a way to formalize the selection process for $\pi_{tgt}$? I understand the novelty of the attack type in isolation, however, the only well-defined πtgt would be min-reward, which is the existing attack paradigm.
> >
> > > For reasons explained above, reward design in Metaworld is task-dependent, $\pi_{tgt}$
> > does not correspond to a reward-minimizing policy.
> >
> > Am I correct in understanding that $\pi_{tgt}$ must be its own pre-defined objective (in the scope of this work)? That is, an environment such as Mujoco-Halfcheetah only has one objective (distance traveled) and thus is ineligible. If this is the case, what occurs when there are two different target objectives? Is there a notion of optimality when selecting $\pi_{tgt}$?

---

> ### Author Response · Authors · 2024-11-28
> **Response to Reviewer uL7G (Part 1)**
>
> We thank you for taking the time to provide us with your thoughtful feedback and consideration. We would like to clarify several points about our position.
>
> ## Q1. Additional Defense Baselines
>
> > To clarify, are the authors stating that [1,2,3] and ATLA-PPO/WocaR-PPO belong to the same group of methodologies?
>
> No, we are not claiming that [1,2,3] and ATLA-PPO/WocaR-PPO belong to **the same methodological group**. Our assertion is specifically that [1,2,3] and ATLA-PPO/WocaR-PPO belong to the same group **in terms of their defense objectives**.
>
> These methods are all reward-based defense approaches that aim to minimize the reduction in the victim's cumulative reward due to attacks. In contrast, our defense method is a behavior-based approach that aims to minimize changes in the victim's behavior resulting from attacks.
>
> **No defense method optimized for the proposed threat model has been proposed so far.**  Thus, I would like to emphasize that comparing the proposed method with such baselines is impossible. Still, we used ATLA-PPO/WocaR-PPO as comparison methods to show that the reward-based defense methods are ineffective against behavior manipulation attacks. For these reasons, we do not consider [1,2,3] to be essential baselines. Reviewer M1ng also has acknowledged the appropriateness of our baseline selection as one of the paper's strengths. However, we agree that including additional baselines could provide a more comprehensive analysis of our defense method.
>
> For this reason, we examined the comparison with [1, 2, 3] as follows. [1] employs an adaptive defense approach, which differs from our defense problem setting. Since the defender in [1] requires additional capabilities that allow it to continuously select the best defensive policy based on the results of the previous episode, which is not allowed for our setting,
>  it would be challenging to implement in our experimental setup. Additionally, [2] has not released their experimental code, making implementation difficult in the short term.
>
> For the comparison with [3], We conducted additional experiments. The results are shown below.
>
> | Adv Task | Methods | Clean Rewards ($\uparrow$) | Best Attack Rewards ($\downarrow$)|
> | --- | --- | --- | --- |
> | window-close | PPO | 4508 $\pm$ 121 | 4505 $\pm$ 65 |
> | | ATLA-PPO | 4169 $\pm$ 467 | 4270 $\pm$ 188 |
> | | PA-ATLA-PPO | 4354 $\pm$ 89 | 2124 $\pm$ 782 |
> | | TDRT-PPO(ours) | 4412 $\pm$ 55 | 482 $\pm$ 3 |
> | window-open | PPO | 4543 $\pm$ 39 | 566 $\pm$ 523 |
> | | ATLA-PPO | 4566 $\pm$ 80 | 586 $\pm$ 649 |
> | | PA-ATLA-PPO | 4332 $\pm$ 109 | 512 $\pm$ 716 |
> | | TDRT-PPO(ours) | 4383 $\pm$ 57 | 254 $\pm$ 214 |
> | drawer-close | PPO | 4714 $\pm$ 16 | 4760 $\pm$ 640 |
> | | ATLA-PPO | 4543 $\pm$ 102 | 4858 $\pm$ 6 |
> | | PA-ATLA-PPO | 4659 $\pm$ 1 | 4865 $\pm$ 3 |
> | | TDRT-PPO(ours) | 4237 $\pm$ 93 | 4860 $\pm$ 4 |
> | drawer-open | PPO | 4868 $\pm$ 6 | 1556 $\pm$ 607 |
> | | ATLA-PPO | 4863 $\pm$ 7 | 1158 $\pm$ 1026|
> | | PA-ATLA-PPO | 4867 $\pm$ 3 | 1054 $\pm$ 219 |
> | | TDRT-PPO(ours) | 4802 $\pm$ 27 | 378 $\pm$ 10 |
>
> While similar trends to ATLA-PPO are observed in many environments, our method demonstrated higher attack resilience than ATLA-PPO in the window-close environment. Our experimental results show that our proposed method achieved the highest robustness across all environments. This reinforces our argument that defending only against untargeted attacks  is insufficient to achieve complete robustness against behavior-targeted attacks.
>
> **We fully acknowledge the relevance of [1,2,3] to our research and have thoroughly discussed these works in our related work section. However, we would like to respectfully emphasize that their objectives differ from those of our proposed method.**
>
> ## Q2. Explanation of WocaR-PPO
>
> > Additionally, a minor note on pedantics: in the paper on lines 458-459, WocaR-PPO is stated to be an adversarial retraining method. To my understanding, the method maximizes a computed lower bound on reward, rather than interacting with an adversary at training time.
>
> We used the term "adversarial training" on lines 458-459 to express the concept of training robust agents by considering worst-case adversaries that minimize rewards. Strictly speaking, as you correctly point out, the method calculates a lower bound on the obtained reward and incorporates it into the objective function.
>
> We apologize for any confusion caused by our limited explanation due to space constraints. We will update this text to be more precise. Thank you for bringing this to our attention.

---

> ### Author Response · Authors · 2024-11-28
> **Response to Reviewer uL7G (Part 2)**
>
> ## Q3. Additional Experimental Results for PA-AD as an Attack Baseline
>
> > However, given that the PGD attacks and Target Reward-Max attacks used in Table 1 each utilize a property in the constraints quoted above, a comparison to white-box PA-AD would make the experiments more thorough, as it is a SOTA method related directly to ATLA-PPO.
>
> Thank you for your valuable feedback.
>
> We agree that the comparison with PA-AD makes our analysis more thorough.　We have conducted experiments with PA-AD. The results are as follows:
>
> | Adv Task | Target Reward (Upper bound of attack) | Random | Targeted PGD | Reward Maximization | PA-AD | BIA-ILfD(ours) | BIA-ILfO(ours)|
> | --- | --- | --- | --- | --- | --- | --- | --- |
> | window-close | 4543 $\pm$ 39 | 947 $\pm$ 529 | 1057 $\pm$ 652 | 4505 $\pm$ 65 | 4255 $\pm$ 300 | 3962 $\pm$ 666 | 4036 $\pm$ 510 |
> | window-open | 4508 $\pm$ 121 | 322 $\pm$ 261 | 296 $\pm$ 117 | 506 $\pm$ 444 | 493 $\pm$ 562 | 566 $\pm$ 523 | 557 $\pm$ 679 |
> | drawer-close | 4868 $\pm$ 6 | 1069 $\pm$ 1585 | 1031 $\pm$ 1437 | 4658 $\pm$ 747 | 3768 $\pm$ 1733 | 4760 $\pm$ 640 | 4626 $\pm$ 791 |
> | drawer-open | 4713 $\pm$ 16 | 841 $\pm$ 357 | 825 $\pm$ 326 | 1499 $\pm$ 536 | 1607 $\pm$ 355 | 1556 $\pm$ 607 | 1445 $\pm$ 610 |
>
> For PA-AD, we conducted experiments setting the objective as maximizing the target reward rather than minimizing the victim's cumulative reward. While the highest attack performance is observed in the drawer-open environment, the overall trends are similar to those of the Reward Maximization Attack.
>
> Although PA-AD demonstrates superior results to SA-RL in untargeted attacks[3], it showed no significant difference from Target Reward Maximization Attack in behavior-targeted attacks. **We hypothesize that environments where all attack methods fail are inherently very challenging to attack.** Therefore, even if PA-AD efficiently designs perturbations, the attack performance does not improve because appropriate false states that would enable successful attacks may not exist in these environments.
>
> ## Q4. The Efficiency of Multi-objective Agent against BIA
>
> > Given that there are several tasks per environment in Metaworld, would a multi-objective agent be something to consider as a robust alternative? I am less familiar with the specifics of multi-objective RL, but it would be an interesting comparison, something to the effect of: $R_{MO}(\cdot) = R(\cdot) - R_{tgt}(\cdot)$
>
> We believe that the multi-objective agent you suggested could potentially enhance robustness against behavior-targeted attacks. Thank you for this insightful suggestion. However, in real-world attacks, adversaries might set attack objectives entirely unrelated to the tasks. In such cases, defenders cannot know what attack objectives or target rewards the adversary will use. Therefore, they cannot utilize reward functions like $R_{MO}(\cdot) = R(\cdot) - R_{tgt}(\cdot)$.
>
> ## Q5. The Objective and Selection of $\pi_{tgt}$
>
> > Am I correct in understanding that $π_{tgt}$ must be its own pre-defined objective (in the scope of this work)?
>
> Yes, $\pi_{tgt}$ follows the adversary's objectives, which are not restricted and can be arbitrary. While we used $\pi_{tgt}$ targeting pre-defined opposite tasks for clear evaluation in our experiments, **the adversary's goals need not be limited to achieving specific tasks.**
>
> > That is, an environment such as Mujoco-Halfcheetah only has one objective (distance traveled) and thus is ineligible.
>
> In the Halfcheetah environment, although the victim's training objective is the distance traveled, **the adversary's objectives can extend beyond this.** For example, they might want to make the agent jump or move alternately to the left and right. In such cases, $\pi_{tgt}$ would be designed according to these adversary objectives.
>
> > If this is the case, what occurs when there are two different target objectives? Is there a notion of optimality when selecting $\pi_{tgt}$?
>
> No, there is no notion of optimality. Even with two target objects, the adversary can freely select $\pi_{tgt}$. For instance, $\pi_{tgt}$ could be designed to achieve either one of the targets or to achieve both targets.
>
> Thank you again for your detailed feedback. This discussion is invaluable in helping us improve our paper. We look forward to your response and continuing this constructive dialogue.

---

> ### Author Response · Authors · 2024-12-02
> **Gentle Reminder**
>
> Dear Reviewer uL7G,
>
> We would like to express our sincere gratitude for your valuable time and effort in reviewing our work.
>
> We are writing to kindly remind you that the discussion period is drawing to a close.
>
> If you have any remaining questions or concerns regarding our paper, we would be most grateful for the opportunity to address them. We are more than happy to provide any clarification you may need.
>
> We fully understand your busy schedule and deeply appreciate your dedication to the review process. Your expertise and insights are invaluable to us.
>
> Thank you again for your continued support and consideration.
>
> Best regards,

---

### Official Review · Reviewer_JMYu · 2024-11-03

**Soundness:** 2
**Presentation:** 2
**Contribution:** 2
**Rating:** 5
**Confidence:** 4

**Summary:**

This paper introduces BIA (Behavior Imitation Attack), a novel attack method that forces a victim RL agent to imitate a specified target policy without requiring environment-specific heuristics or full access to the victim's policy. Rather than attempting to match actions at each individual step, BIA aims to achieve overall behavioral alignment, making it effective even when perfect action matching is impossible due to limited adversarial intervention capabilities. Additionally, the paper presents TDRT (Time-Discounted Regularization Training), the first defense framework specifically designed for behavior-targeted attacks, which applies stronger regularization in early trajectory stages and demonstrates improved robustness-performance trade-offs compared to existing untargeted defense methods.

**Strengths:**

1. Shifts from action matching to behavioral imitation, enabling effective attacks under limited intervention.
2. Identifies the critical role of early-trajectory stability and action-state sensitivity in defense against behavior targeted attack
3. Provides theoretical analysis in both attack and defense methods.

**Weaknesses:**

See questions

**Questions:**

1. The nearly identical performance between BIA-ILfD and BIA-ILfO contradicts the established understanding that ILfO is generally more challenging than ILfD. This observation in very easy MetaWorld tasks significantly weakens the paper's empirical validation.
2. The paper fails to analyze how the quantity and quality of demonstrations affect attack performance. The use of 20 demonstrations for these MetaWorld tasks seems excessive.
3. No discussion on whether expert-level demonstrations are necessary. The assumption of having access to expert demonstrations may severely limit practical applicability.
4. Lack of limitation and efficiency analysis for the proposed attack and defences
5. Overlooks recent related work [1] about RL defences

[1] Game-Theoretic Robust Reinforcement Learning Handles Temporally-Coupled Perturbations, Liang et al, ICLR 2024

---

> ### Author Response · Authors · 2024-11-18
> **Response to Reviewer JMYu (Part 1)**
>
> We thank the reviewer for their valuable feedback. Below, we provide clarifications:
>
> ## Q1. Performance Similarity Between ILfD and ILfO
>
> > Q1. The nearly identical performance between BIA-ILfD and BIA-ILfO contradicts the established understanding that ILfO is generally more challenging than ILfD. This observation in very easy MetaWorld tasks significantly weakens the paper's empirical validation.
>
> We believe that the similar performance of ILfO and ILfD is due to the deterministic nature of the transition probability in the MetaWorld tasks. The control tasks used in the experiments in this paper simulate the physical world, so the state after state transition is almost uniquely determined. In such a situation, ILfO and ILfD are almost identical. This may have resulted in minimal differences between the information accessible by ILfO and ILfD.
>
> We expect that in environments where the transition probability is more stochastic and the state transitions of actions are more random, the information about the expert's action choices becomes more crucial, leading to a more noticeable performance gap between ILfO and ILfD. For example, recommendation tasks and interactive control, as shown in Response 1 to Reviewer M1ng, state transitions are more stochastic.
>
> ## Q2. Demonstration Requirements Analysis
>
> > Q2. The paper fails to analyze how the quantity and quality of demonstrations affect attack performance. The use of 20 demonstrations for these MetaWorld tasks seems excessive.
>
> We have conducted additional experiments and analysis on the relationship between demonstrations and attack performance and added them to Section F.2. Due to space limitations, we present only the BIA-ILfD results here. Additional experimental results for the ILfO setting are provided in Section F.2.
>
> ### BIA-ILfD Attack Performance Results
>
> | env | 1 demo | 4 demo | 8 demo | 12 demo | 16 demo | 20 demo |
> |---|---|---|---|---|---|---|
> | window close | 2616 $\pm$ 2278 | 3727 $\pm$ 598 | 3507 $\pm$ 1066 | 4083 $\pm$ 523 | 4307 $\pm$ 505 | 3962 $\pm$ 666 |
> | window open | 438 $\pm$ 283 | 552 $\pm$ 679 | 629 $\pm$ 632 | 627 $\pm$ 599 | 637 $\pm$ 605 | 566 $\pm$ 523 |
> | drawer close | 3454 $\pm$ 709 | 4309 $\pm$ 209 | 4540 $\pm$ 405 | 4627 $\pm$ 475 | 4448 $\pm$ 314 | 4760 $\pm$ 640 |
> | drawer open | 1179 $\pm$ 625 | 1593 $\pm$ 431 | 1559 $\pm$ 373 | 1545 $\pm$ 556 | 1462 $\pm$ 599 | 1556 $\pm$ 607 |
>
> As the reviewer correctly pointed out, our experiments confirmed that 20 demonstration episodes are excessive for MetaWorld tasks. The results showed that performance was maintained with as few as four demonstration episodes, with negligible degradation in performance.
>
> When the demonstrations were reduced to just one episode, we observed a significant decline in performance. This can be attributed to the stochastic initial state distributions in MetaWorld tasks. Specifically, with such a limited number of demonstrations, the discriminator struggles to develop sufficient generalization capabilities across various initial states.

---

> ### Author Response · Authors · 2024-11-18
> **Response to Reviewer JMYu (Part 2)**
>
> ## Q3. Expert Demonstration Requirements and Practicality
>
> > Q3. No discussion on whether expert-level demonstrations are necessary. The assumption of having access to expert demonstrations may severely limit practical applicability.
>
> Thank you for raising this important point about the need for expert demonstrations. We would like to clarify our position on this issue.
>
> Unlike untargeted attacks, **targeted attacks inherently require information that indicates the adversary's intended behaviors**. This is similar to targeted attacks on classifiers, where one must specify the label to be forced after the attack.
> Since the adversary's goal is to make the victim perform a specific behavior, the adversary needs to specify that behavior. In our research, this is referred to as an expert demonstration.
>
> **Please note that generating an expert demonstration is not necessarily expensive compared to designing reward functions or learning expert policies**. For example, in an autonomous driving scenario where an adversary wants to make the victim pass through a particular intersection, the adversary can create expert demonstrations simply by driving through the intersection several times and recording the trajectories.
>
> ## Q4. Limitations and Efficiency Analysis
>
> > Q4. Lack of limitation and efficiency analysis for the proposed attack and defences
>
> We have added discussions on limitations to the conclusion section and discussions on efficiency to Section E. Additionally, we plan to include experiments analyzing the relationship between attack budget and attack performance as a more detailed efficiency analysis of our proposed method during the period of the rebuttal.
>
> ## Q5. Related Work Updates
>
> > Q5. Overlooks recent related work [1] about RL defences
>
> Thank you for pointing out the related work.
> We have added a discussion of this research to our related work section.

---

> ### Author Response · Authors · 2024-11-25
> **Response to Reviewer JMYu (Part 3)**
>
> Dear Reviewer JMYu,
>
> We have conducted additional experiments investigating the relationship between attack budget and attack performance and updated our draft. The detailed results and analysis can be found in Section F.3.
>
> Thank you very much for your valuable feedback. We have done our best to address your concerns. We would appreciate any additional feedback to help us improve the paper.

---

> > ### Comment · Reviewer_JMYu · 2024-11-27
> > **Response to authors**
> >
> > Thank you for your response. I think all my concerns have been adequately addressed. Since I consider this paper to be a borderline case, I will discuss with the other reviewers and Area Chair before making the final decision.

---

> > > ### Author Response · Authors · 2024-11-28
> > > **Thank you for your valuable comments**
> > >
> > > ### We sincerely appreciate the time and effort you have taken to review our work.
> > >
> > > We are truly glad that all of your concerns have been adequately addressed. With this in mind, we humbly believe that a score adjustment toward acceptance might now be appropriate. **If there are any remaining concerns you feel may outweigh the reasons for acceptance, including those raised by other reviewers, we would be most grateful for the opportunity to discuss them further.**
> > >
> > > We would like to thank you once again for taking the time to provide us with your thoughtful feedback and consideration.

---

### Official Review · Reviewer_M1ng · 2024-11-05

**Soundness:** 3
**Presentation:** 3
**Contribution:** 3
**Rating:** 6
**Confidence:** 4

**Summary:**

The paper studies the problem of deep RL under poisoning attack. It considers an adversary to manipulate the behavior of a policy used by the learning agent during testing time. It develops an efficient attack and defense in this setting and provides the theoretical insight behind its methods. The performance of its methods is verified with abundant experiments, and the experiment results seem promising.

**Strengths:**

1. This paper studies an interesting and important problem: adversarial attacks in DRL and their defense

2. This paper provides a complete story that gives both an attack and a defense. In addition, for the attack part, they are the first black-box attack in their targeted testing time attack setting.

3. The selection of the baselines is reasonable, and the experiment results look very strong. Both the attack and defense perform significantly better than their baselines.

4. Considering the promising results, their methods seem simple and easy to implement, which makes them more practical.

**Weaknesses:**

1. The explanation of the motivation is not enough. This work only explains an example of autonomous driving. It should also explain why the attack is realistic: how this attack happens in the real world and what bad things will happen under the attack. It is even better to find evidence showing that behavior manipulation attacks already threaten our world.

2. An explanation of the challenges is missing. The problem studied in this problem should not be easy, but it is beneficial to explicitly clarify exactly why the problem is hard to solve and what the challenges are.

3. This work's theoretical analysis is weak in that it doesn't provide any guarantees on their defense algorithm. Is there a chance that some guarantees can be provided under certain assumptions?

4. It is difficult to digest the result that the PGD attack performs almost identically to the random attack. I want to see more details on how the PGD attack is implemented and how much effort has been spent on tuning it. This baseline ought to be strong, but now it seems trivial. I think a more detailed investigation is necessary rather than a simple explanation.

5. It took me a while to find that the attack works during the testing time (no training happens during the attack). I think this needs to be clarified. It is also beneficial to discuss the related works on the training time DRL attacks to make the related work section complete, such as [1] and [2].

[1]: Sun, Yanchao, Da Huo, and Furong Huang. "Vulnerability-aware poisoning mechanism for online rl with unknown dynamics." arXiv preprint arXiv:2009.00774 (2020).

[2]: Xu, Yinglun, Qi Zeng, and Gagandeep Singh. "Efficient Reward Poisoning Attacks on Online Deep Reinforcement Learning." Transactions on Machine Learning Research.

**Questions:**

See Weakness

---

> ### Author Response · Authors · 2024-11-18
> **Response to Reviewer M1ng (Part 1)**
>
> We thank the reviewer for their valuable feedback. Below, we provide clarifications:
>
> ## W1. Motivation and Real-World Threats
>
> > W1. The explanation of the motivation is not enough. This work only explains an example of autonomous driving. It should also explain why the attack is realistic: how this attack happens in the real world and what bad things will happen under the attack. It is even better to find evidence showing that behavior manipulation attacks already threaten our world.
>
> The behavioral manipulation attacks we propose are a threat in situations where the adversary's goal is not to reduce the victim's reward, but to manipulate the victim's behavior. As real-world examples where behavior manipulation attacks could pose significant threats, we would like to highlight **recommendation systems** and **dialogue control** using large language models (LLMs).
>
> **Recommender system:** Recently, there has been active research on applying reinforcement learning to recommendation systems[1,2]. In a recommendation system that uses reinforcement learning, recommendations are continuously provided that increase the user's long-term satisfaction, using information that is continuously observed from the user's behavior. In this situation, an adversary aiming to reduce the reward would simply aim to provide recommendations that would lower the user's satisfaction. On the other hand, in the proposed behavior manipulation attack, the adversary aims to lure the user into a specific situation, for example, an attack that manipulates the recommendations so that the user selects or purchases specific items.
>
> **Dialogue Control:** In addition, reinforcement learning is used in language models to design dialogues with long-term goals[3]. For example, it can be used to improve the consistency of responses in conversations with users, maximize satisfaction based on the context of the conversation, and improve the success rate of tasks with specific goals (such as making restaurant reservations, shopping assistants, and customer support). In this context, our behavioral manipulation attack can be an attack that makes users perform specific actions without them noticing, such as making users book a specific restaurant.
>
> ## W2. Problem Complexity and Technical Challenges
>
> > W2. An explanation of the challenges is missing. The problem studied in this problem should not be easy, but it is beneficial to explicitly clarify exactly why the problem is hard to solve and what the challenges are.
>
> The main challenge in behavioral manipulation attacks and defense is that they are executed in **multi-step rather than single-step processes**.
>
> For attack, while PGD attacks can successfully induce desired behaviors in a single step, they become ineffective when extended to multiple steps, as shown in our paper with the results in Table 1. To overcome difficulty, we propose a method that learns the adversarial policy through imitation learning in Section 5. In imitation learning, the discriminator's reward serves as an appropriate measure for multi-step behaviors.
>
> For defenses, while our goal is to minimize changes in the victim's overall behavior, directly computing these overall behavioral changes presents significant technical challenges. To address this challenge, we prove that the difference in state-action distributions across multiple steps can be bounded by the difference in action distributions in a single step in Section 6.
>
>
> [1] Chen, Xiaocong, et al. "Deep reinforcement learning in recommender systems: A survey and new perspectives." Knowledge-Based Systems 264 (2023): 110335.
>
> [2] Fu, Mingsheng, et al. "A deep reinforcement learning recommender system with multiple policies for recommendations." IEEE Transactions on Industrial Informatics 19.2 (2022): 2049-2061.
>
> [3] Deborah Cohen and Craig Boutilier. "Using reinforcement learning for dynamic planning in open-ended conversations," https://research.google/blog/using-reinforcement-learning-for-dynamic-planning-in-open-ended-conversations/

---

> ### Author Response · Authors · 2024-11-18
> **Response to Reviewer M1ng (Part 2)**
>
> ## W3. Theoretical Guarantees on the defense algorithm
>
> > W3. This work's theoretical analysis is weak in that it doesn't provide any guarantees on their defense algorithm. Is there a chance that some guarantees can be provided under certain assumptions?
>
> Thank you for the insightful comment regarding theoretical guarantees.
>
> Please understand that this is **the first paper to propose a defense method against behavioral manipulation attack**, and that theoretical guarantees for defense remain as a future issue.
>
> We believe that certified defense against behavioral changes could be achieved by adapting randomized smoothing similar to the certified defense method for untargeted attacks[4]. This would be a promising direction for future research to establish theoretical guarantees under appropriate assumptions.
>
> ## W4. PGD Attack Performance Analysis
>
> > W4. It is difficult to digest the result that the PGD attack performs almost identically to the random attack. I want to see more details on how the PGD attack is implemented and how much effort has been spent on tuning it. This baseline ought to be strong, but now it seems trivial. I think a more detailed investigation is necessary rather than a simple explanation.
>
> Thank you for raising this important point about the PGD attack baseline.
>
> We added further experiments and analysis of PGD attacks, as well as implementation details, to Section G.2.
>
> To summarize, when the attack budget $\epsilon$ was set to very large values, such as 1.0 or higher, we confirmed that the targeted PGD attack works successfully. This is because it becomes possible that false states that completely mimic the target policy's actions can be found at each step if such large attack budgets are given at any time. However, when the attack budget is limited, false states that can mimic the target policy are not found, resulting in overall behavior that is completely different from the target policy and making the attack ineffective.
>
> ## W5. Attack Timing and Related Work Clarification
>
> > W5. It took me a while to find that the attack works during the testing time (no training happens during the attack). I think this needs to be clarified. It is also beneficial to discuss the related works on the training time DRL attacks to make the related work section complete, such as [1] and [2].
>
> Thank you for pointing out the need for clarity regarding the attack timing and the suggestion to expand our discussion of related works.
>
> We have updated the introduction section to emphasize the attack timing.
> While we define the attack as occurring during testing time in the threat model, we agree that this point was not immediately clear. Additionally, we have expanded the related work section to include discussions on poisoning attacks, which are performed during the victim's training time.
>
>
> [4] Wu, Fan, et al. "CROP: CERTIFYING ROBUST POLICIES FOR REINFORCEMENT LEARNING THROUGH FUNCTIONAL SMOOTHING." 10th International Conference on Learning Representations, ICLR 2022.

---

> ### Comment · Reviewer_M1ng · 2024-11-29
>
> Thanks for your response. I have raised my score.

---

> > ### Author Response · Authors · 2024-12-02
> > **Thank You for Your Positive Evaluation**
> >
> > Dear Reviewer M1ng,
> >
> > We are deeply grateful for your positive evaluation of our revised manuscript. Your thoughtful feedback and constructive comments throughout the review process have significantly improved the quality of our work.
> >
> > We truly appreciate the time and expertise you have dedicated to reviewing our manuscript. Your insights have been invaluable.
> >
> > Thank you again for your support and consideration.

---

### Meta-Review · Area_Chair_aFp2 · 2024-12-21

**Metareview:**

The reviewers agree that the paper addresses an interesting and novel problem, but the contribution falls short in terms of motivation, theoretical rigor, and empirical strength. Specific concerns include the limited applicability of the proposed methods, inadequate baseline comparisons, and a lack of compelling real-world examples to justify the problem's significance. Despite revisions during the discussion phase, these critical issues remain unresolved.

Thus, I recommend rejection.

**Additional Comments On Reviewer Discussion:**

NA

---

### Decision · Program_Chairs · 2025-01-22

Reject